



# Understanding Cluster Wake-Induced Energy Losses off the U.S. East Coast

Geng Xia[1], Mike Optis[2], Georgios Deskos[1], Michael Sinner[1], Daniel Mulas Hernando[1], Julie K. Lundquist[1,3], Andrew Kumler[1,4], Miguel Sanchez Gomez[1], Paul Fleming[1], and Walter Musial[1]

[1]National Renewable Energy Laboratory, Golden, CO, USA
[2]Veer Renewables, Courtenay, British Columbia, Canada
[3]Johns Hopkins University, Baltimore, MD, USA
[4]University of Colorado, Boulder, CO, USA

**Correspondence:** Geng Xia (geng.xia@nrel.gov) and Michael Sinner (michael.sinner@nrel.gov)

**Abstract.** This study seeks to advance our understanding of energy losses caused by wind farm cluster wakes off the U.S. East Coast by utilizing advanced numerical models in conjunction with real-world, available data on existing and planned offshore wind sites. To this end, we have run simulations of existing and planned U.S. offshore wind lease areas using a typical-meteorological-year approach with a GPU-based Weather Research and Forecasting (WRF) model, where lease area layouts are generated based on most up-to-date project capacity information for each individual lease areas. To evaluate wake losses, we use an energy-loss-based definition of "wake shadow", as opposed to the traditional wind speed deficit assessment. A key insight from this study is that large wind speed deficits do not necessarily translate into significant energy losses. In addition, our results indicate that the conventional wind speed deficit method may underestimate the size of the wake area by up to 30% compared to the proposed energy loss approach. These findings highlight the need to consider both wind speed deficits and energy losses when evaluating the wake effects of offshore wind farms and assessing future offshore wind development.

## 1 Introduction

With its reliable 100-meter wind resource ($>8.5\,\mathrm{m\,s^{-1}}$), shallow water depths ($<60\,\mathrm{m}$), and close proximity to large population centers, the U.S. Atlantic Outer Continental Shelf (OCS) is an ideal place to harvest offshore wind energy (Bodini et al., 2019; Musial et al., 2016). Currently, there are 30 offshore lease areas over this region in various stages of development (Bureau of Ocean Energy Management, 2024), 26 of which are examined in this study. However, if all these projects advance to installation and commissioning, concerns regarding wake-induced energy losses from upstream and neighboring farms have been raised (Dörenkämper et al., 2014; Lundquist et al., 2019; Golbazi et al., 2022; Pryor and Barthelmie, 2024b, a; Rosencrans et al., 2024). Moreover, the establishment of new wind farms on future lease areas could further impact the energy-generating potential of existing leases, highlighting the need for more informed and transparent site selection. To this end, a robust assessment of the wake effects is critical to ensure the sustainable future growth of offshore wind and address an industry-wide problem.



Wake-induced energy losses have long been recognized as a critical challenge for offshore wind energy (Lissaman, 1979; Ainslie, 1988; Rados et al., 2001). Initially, research focused on the effects within a single wind farm (intra-array) (Barthelmie et al., 2009) and between closely spaced wind farms (deep-array) (Nygaard, 2014). More recently, the emphasis has shifted to cluster wake effects (Platis et al., 2018; Lundquist et al., 2019; Cañadillas et al., 2022; Schneemann et al., 2020), in which wakes from one wind farm can extend far into the surrounding area, impacting other offshore wind farms in the far field (Platis et al., 2018). Research on cluster wakes has predominantly concentrated on the North Sea, driven by the region's increasingly dense arrangement of offshore wind farms and future climate change considerations (Akhtar et al., 2021; Warder and Piggott, 2025). The extensive size and close spacing of wind farms influence not only individual downstream turbines but also entire neighboring downstream facilities (Cañadillas et al., 2020; Ahsbahs et al., 2020), potentially lowering the overall capacity factor by 20% or more (Akhtar et al., 2021).

Atmospheric modeling has been at the forefront of cluster wake research (Fischereit et al., 2022a). In particular, mesoscale numerical weather prediction models have become widely used for assessing long-term wake effects thanks to their ability to capture the atmospheric conditions (e.g., stability) influencing wake evolution at a reasonable computational cost (Fitch et al., 2012). Validating these models with field measurements and comparing wake estimates across different model fidelities have been essential for quantifying uncertainty and revealing current model limitations (Dörenkämper et al., 2015; Lee and Lundquist, 2017; Tomaszewski and Lundquist, 2020; Siedersleben et al., 2018b, a, 2020; Fischereit et al., 2022b; Maas, 2023; Ali et al., 2023a). For example, Cañadillas et al. (2022) collected both airborne and scanning wind lidar data and compared them to a mesoscale model with wind farm parameterization. Under neutral and unstable conditions, the model and measurements agreed within about 2% for wind speed, but stable conditions introduced the largest discrepancies. Similarly, Sanchez Gomez et al. (2024) compared operational data from the Westermost Rough Offshore Wind Farm, subjected to partial or full wakes from the Humber Gateway array, against high-resolution mesoscale simulations. They found that while mesoscale models with wind farm parameterizations generally represent the influence of cluster wakes on the front-row turbines, they often fail to capture cluster wake effects across the entire farm, likely due to misrepresentation of internal wake dynamics.

Recent studies have also focused on offshore wind development off the U.S. East Coast, offering insights into how large-scale wind farm wakes interact with the region's distinct atmospheric conditions. Pryor et al. (2021a) presented the first quantitative analysis of how offshore wind energy lease areas along the U.S. East Coast influence power production and wake formation, focusing on selected short periods. Their study considered capacity densities between $2.1\,\mathrm{MW/km^2}$ and $4.3\,\mathrm{MW/km^2}$ and found that energy output may diminish by approximately one-third due to the wakes produced by upwind turbines and wind farms. Under some atmospheric conditions, extensive wind farm wakes can extend as far as $90\,\mathrm{km}$ beyond the largest lease areas, and the frequency-weighted area experiencing a 5% velocity deficit can be 2.6 times the size of the original lease area. Based on their simulations, they introduced scaling rules that relate wake impacts from large offshore wind developments to prevailing meteorological conditions and the density of installed wind turbines. Rosencrans et al. (2024) examined a complete annual cycle of wake impacts on the East Coast. The study demonstrates that offshore wind farm wakes can reduce power output by approximately 34% to 38%, with the strongest wakes extending up to $55\,\mathrm{km}$ during summertime stable atmospheric conditions. They also pointed out that internal wakes may contribute more to energy losses than external wakes, and the overall



wake impact is influenced by nonlinear processes. By varying the wake-added turbulent kinetic energy (TKE) amount, they quantified uncertainty in the yielded energy output variability, which was estimated at 3.8%. Pryor and Barthelmie (2024a) revisited their 2021 study, projecting that the annual energy production from current offshore wind energy leases ranges from 139 to 173 TWh/yr and demonstrating out that the combined intra- and inter-array wake effects can extend well beyond existing lease areas, reducing the annual energy production by up to 49 TWh/yr. They also considered the fraction of U.S. East Coast resource that remains available after considering wake effects and competing ocean uses (shipping activity, distance to coast, etc.). They suggested that there are more than $40\,000\,\mathrm{km}^2$ available in shallow water for offshore wind development. We note that marine spatial planning is a thorough process managed by the Bureau of Ocean Energy Management (BOEM), which is responsible for overseeing renewable energy development on the outer continental shelf. As a result, any assessment, including this study, may not fully reflect the actual availability of resources.

All of the studies mentioned above used numerical weather prediction models equipped with a wind farm parameterization (Fitch et al., 2012; Volker et al., 2015) to examine the impact of wakes over an offshore wind farm. To complement and expand on the findings of previous studies, in this study we provide a detailed analysis of the wake-induced energy losses along the U.S. East Coast. In contrast to previous studies, we focus on energy losses, rather than velocity deficits, when assessing the extent of wind farm cluster wakes. We also differ from previous studies in our approach for generating wind farm layouts based on up-to-date information from BOEM (BOEM, 2024) to better approximate the installed density capacity of individual lease areas. We find that using the more accurate heterogeneous capacity density (CD) can affect the extent of the cluster wakes and the magnitude of the energy deficit for downstream wind farms. Moreover, we adopt the typical-meteorological-year (TMY) method in contrast to single year-long simulations (Rosencrans et al., 2024) or multiple short-period simulations (Pryor et al., 2021b; Pryor and Barthelmie, 2024a) in an effort to capture seasonal and inter-annual variability in wake-induced energy loss assessment.

## 2 Simulation Methodology

This section introduces the simulation methodology used to assess wind farm cluster wakes at lease areas along the U.S. East Coast. We first describe the Weather Research and Forecasting (WRF) model used, before describing the wind farm characterization and the approach used to generate wind farm layouts within the specified lease areas.

### 2.1 WRF model setup and experiment design

The version of the WRF model (Skamarock et al., 2019) used here was developed by TempoQuest, a company based in Boulder, CO, USA, and was built to run on accelerated graphical processing units (GPUs) (Veer, 2023). The GPU-based WRF version, called AceCAST, allows for rapid acceleration of WRF simulations and was used explicitly in this project to leverage the National Renewable Energy Laboratory's (NREL) modern NVIDIA H100 GPUs on its Kestrel high-performance computing platform. AceCAST encompasses a set of refactored common WRF physics and dynamics modules and namelist options with



| Shortwave & longwave radiation | Rapid radiative transfer model for global climate models (RRTMG; Iacono et al., 2008) |
|---|---|
| Planetary boundary layer | Mellor-Yamada-Nakanishi-Niino (MYNN; Nakanishi and Niino, 2004, 2009) |
| Surface layer | Revised MM5 Monin-Obukhov scheme (Jiménez et al., 2012) |
| Land surface model | Noah land surface model (Chen and Dudhia, 2001) |
| Cumulus | Kain-Fritsch scheme (Kain and Fritsch, 1990; Kain, 2004) |
| Microphysics | Single-moment 3-class (WSM3) simple ice scheme (Hong et al., 2004) |
| Wind farm parameterizations | Fitch scheme (Fitch et al., 2012); TKE generation factor = 1.0 |

**Table 1.** Summary of physics parameterizations in the WRF model.

NVIDIA CUDA or OpenACC GPU programming techniques and is built to function as an interchangable replacement for existing WRF configurations.

In October 2023, Veer Renewables conducted a validation study of AceCAST against the traditional CPU-based WRF model. Using a full-year simulation focused on the U.S. offshore Atlantic wind energy areas, the study found near-perfect agreement between AceCAST version 3.1 and the equivalent WRF version 4.2.2 (Veer, 2023). Based on the results of this study (summarized in Appendix A), the GPU-based equivalent to WRF 4.5.1 is used for conducting our simulations.

As in Pryor et al. (2021b), simulations are performed with three one-way nested domains (Figure 1), focusing on the wind

lease areas over the U.S. East Coast with a spatial resolution of 9 km, 3 km, and 1 km, respectively. The European Centre for Medium-Range Weather Forecasts Version 5 (ERA5) reanalysis data (Hersbach et al., 2020) provide the initial and boundary conditions for the simulation. The physics parameterizations used in this study are summarized in Table 1. In total, the model employs 52 vertical levels, with finer resolution (15 levels) concentrated in the lowest 200 meters above the ground to better capture the interactions between the wind farms and the surface layer (Tomaszewski and Lundquist, 2020).



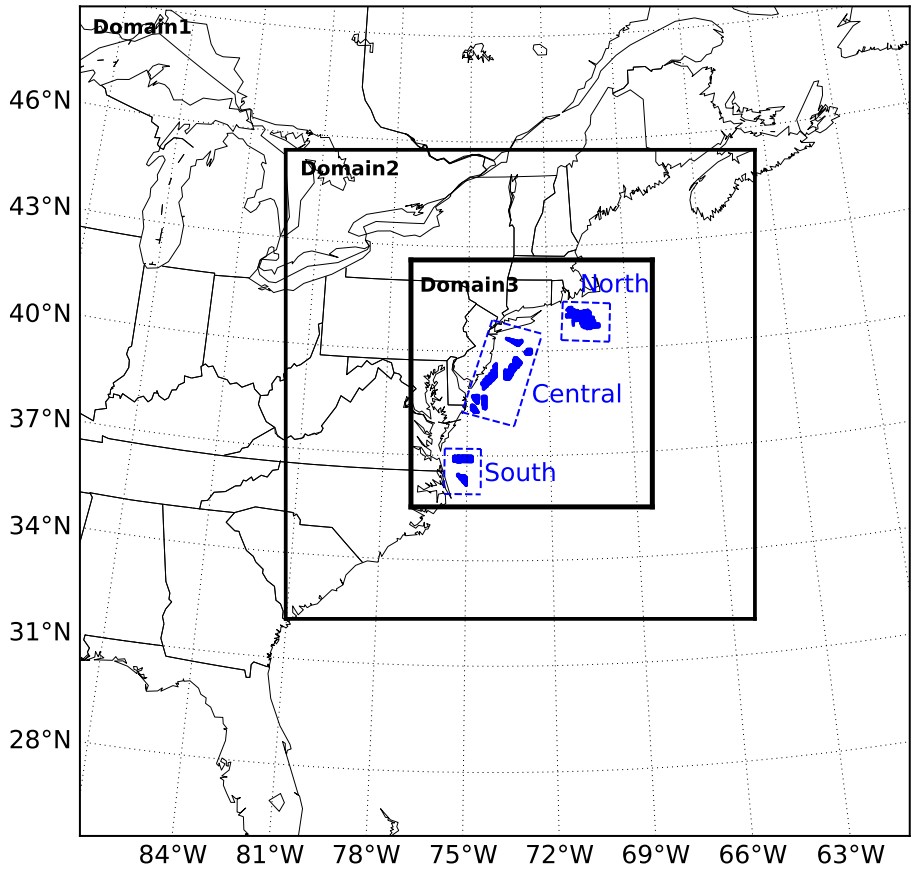

**Figure 1.** Domain design for the WRF simulation. The shaded areas represent the locations of the wind turbines. Domain 1 has 9-km horizontal resolution, Domain 2 has 3-km horizontal resolution, and Domain 3 has 1-km horizontal resolution. The three blue boxes indicate distinct offshore wind lease areas—north, central, and south—used for regional analysis.

A full-year simulation is run in this study. However, rather than running the simulation over a continuous year, we construct a TMY (Knight et al., 1991; Kambezidis et al., 2020; Ren et al., 2021) that is representative of 24 years of atmospheric conditions (Table 2). To construct the TMY, the following steps are performed:

1. At the center coordinate of the model domain, we download 100-m wind speed, 100-m wind direction, and 2-m temperature timeseries data from the ERA5 reanalysis over the period from January 2000 to December 2024.

2. For each calendar month, we compute the Wasserstein metric (also called the Earth mover's distance) between the long-term distribution of the 100-m wind speed and that for each year. We then repeat for the 100-m wind direction and 2-m temperature data.





3. For each atmospheric variable, we normalize the Wasserstein metric by using the mean of all years, allowing comparability of the metrics between atmospheric variables.

4. For each year, we sum the normalized metrics across each atmospheric variable. The year with the lowest summed metric is the one selected in the TMY.

5. Repeat for each calendar month.

| Year | Month |
| --- | --- |
| January | 2008 |
| February | 2001 |
| March | 2007 |
| April | 2013 |
| May | 2010 |
| June | 2020 |
| July | 2017 |
| August | 2006 |
| September | 2019 |
| October | 2013 |
| November | 2021 |
| December | 2004 |

**Table 2.** Summary of the months constructed for the TMY used in this study.

By building the TMY using full calendar months—in contrast to more meticulous TMY approaches where individual days might be selected—it is ensured that extreme events are not excluded. By considering wind speed, wind direction, and 2-m

temperature (related to air density), a more complete picture of the meteorological situation is created. Figure 2 examines the efficacy of this TMY approach by contrasting the full TMY distributions against the 20-year long-term ERA5 data. It is evident that the TMY approach accurately captures the long-term wind climatology from ERA5 and is therefore appropriate for use in this study.

The wind farm parameterization of Fitch et al. (2012), with the bug fixes of Archer et al. (2020), is the default wind farm

parameterization in the WRF model and is used in this study. It represents the effect of a wind turbine by introducing a momentum sink term and a TKE source term into the model layer where wind turbines are located. The following equations show the effect of the wind farm parametrization on 1) horizontal wind speed, 2) power production, and 3) turbulence generation:

$$\frac{\delta |V|_{ijk}}{\delta t} = -\frac{N_{ij} C_T(|V|_{ijk})|V|_{ijk}^2 A_{ijk}}{2(z_{k+1} - z_k)} \qquad (1)$$



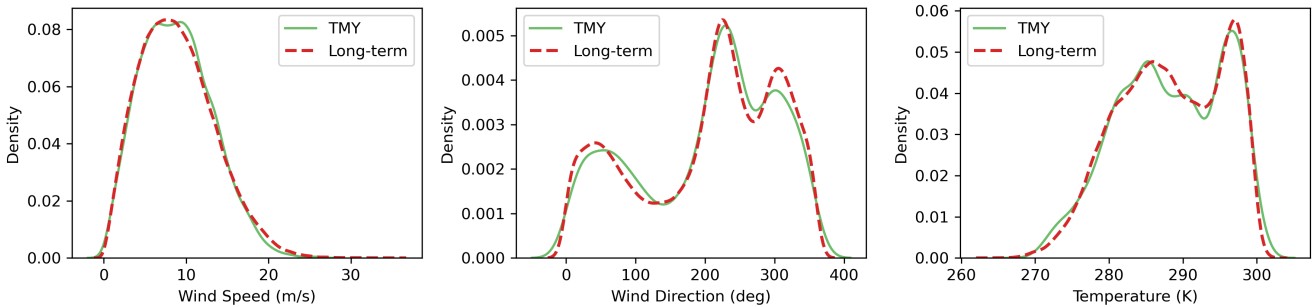

**Figure 2.** Probability distributions of 100-m wind speed (a), 100-m wind direction (b), and 2-m temperature (c) at the model domain center: comparison of TMY approach (green) and 20-year ERA5 data (red-dashed).

$$\frac{\delta P_{ijk}}{\delta t} = \frac{N_{ij} C_P(|V|_{ijk}) |V|_{ijk}^3 A_{ijk}}{2(z_{k+1} - z_k)} \tag{2}$$

$$\frac{\delta \mathrm{TKE}_{ijk}}{\delta t} = \frac{N_{ij} C_{\mathrm{TKE}}(|V|_{ijk}) |V|_{ijk}^3 A_{ijk}}{2(z_{k+1} - z_k)} \tag{3}$$

where $i$, $j$, and $k$ denote Cartesian model coordinates. The thrust coefficient, $C_T(|V|_{ijk})$, varies with wind speed, and $|V|$ represents the wind speed magnitude at the turbine's hub height. The rotor-swept area is denoted as $A_{ijk}$. The turbine number density within a grid cell $ij$ is represented as $N_{ij}$. The power coefficient, $C_P(|V|_{ijk})$, is also dependent on wind speed. The height of the vertical model level is indicated by $z_k$, and $C_{\mathrm{TKE}}$ approximates the fraction of energy converted to TKE. Several modifications to the Fitch scheme (Redfern et al., 2019) and other wind farm parameterizations (Abkar and Porté-Agel, 2015; Volker et al., 2015; Pan and Archer, 2018) have been developed in recent years, as reviewed by Fischereit et al. (2022a). Ali et al. (2023b) provide an in-depth comparison between these wind farm parameterizations and conclude that inclusion of the turbulence source term is critical for an improved prediction of near-surface variables, as shown in comparison of mesoscale simulations to large-eddy simulations of Vanderwende et al. (2016). A development to correct for the blockage effect on upwind wind speed (Vollmer et al., 2024) was introduced to the Fitch parameterization after the beginning of this project and so was not implemented in the present study.

Wind turbines are represented in the WRF simulations by using their thrust coefficient and power as a function of wind speed (see Figure 3). The thrust coefficient is a dimensionless quantity representing the thrust force applied by the turbine on the flow at a given wind speed (Burton et al., 2011, ch. 3.2.4). Three turbine power ratings—11 MW, 13 MW and 15 MW—are selected for this study. The 15-MW turbine coefficients, which make up the bulk of the turbines, are based on the International Energy Agency's (IEA) 15-MW reference turbine (Gaertner et al., 2020) with 20% peak shaving (discussed in the following paragraph). The 11-MW turbine coefficients are scaled-down versions of the IEA 15-MW turbine, using turbine information and technology assumptions from Beiter et al. (2020), combined with the actuator disk theory from Burton et al.

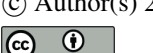

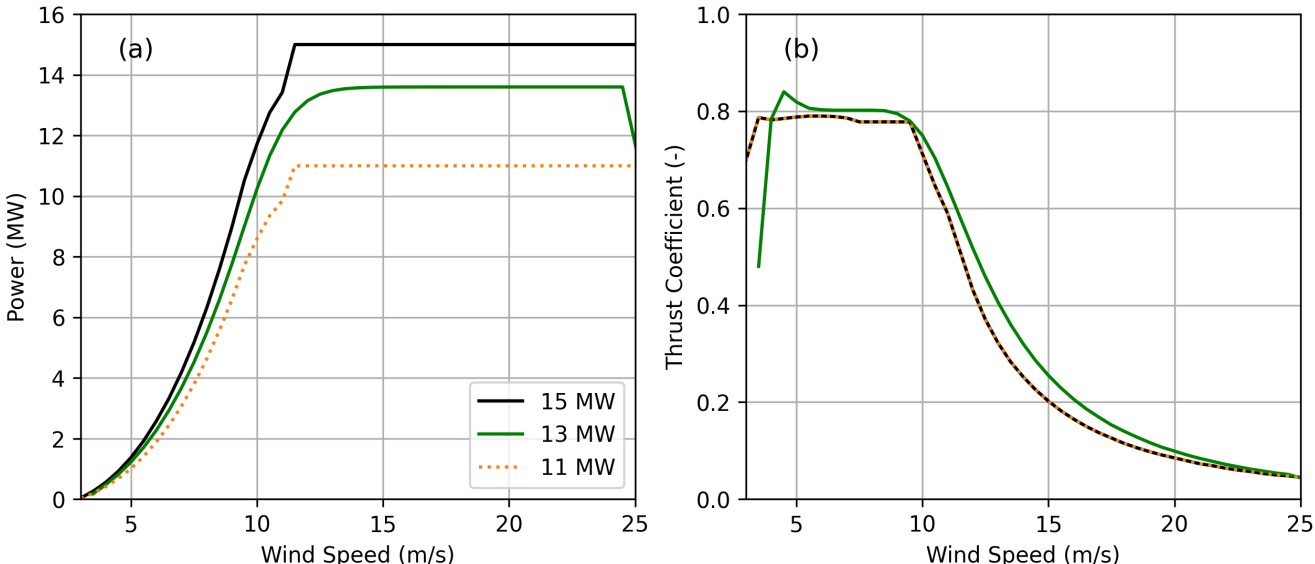

**Figure 3.** Power (a) and thrust coefficient (b) curves for the 11-, 13-, and 15-MW turbines used in this study. The hub height and rotor diameter are 133 m and 205.5 m for the 11-MW turbine, 138 m and 220 m for the 13-MW turbine, and 150 m and 240 m for the 15-MW turbine.

(2011), resulting in the power and thrust coefficient curves in Figure 3. The 13-MW turbine information was obtained from
145 Det Norske Veritas in April 2024 and left unmodified in this study. Note that this turbine's rated power is in fact 13.6 MW; however, for simplicity and consistency we refer to it as "13 MW" and will use it in lease areas where an approximate turbine rating of 13 MW is assumed (see Section 2.2).

Peak shaving operation, where a wind turbine is intentionally operated off of its maximum-power-generating control settings near the rated wind speed to lessen structural loads, is increasingly common for large offshore wind turbines (Peeringa et al.,
2011; Hansen and Henriksen, 2013). To incorporate peak shaving into the simulation, the thrust curve of the IEA 15-MW turbine is altered to achieve a 20% reduction in the peak thrust compared to the nominal thrust curve defined by Gaertner et al. (2020) (see especially Figure 3-1 therein), chosen as a reasonable compromise between load reduction and power loss (Vanelli et al., 2022; Tian et al., 2023). This alteration gives the "trimmed corner" appearance of the thrust curve of the IEA 15-MW turbine between roughly 9 and 12 m/s in Figure 3(b). An approximation of the resulting effect on the power curve based
on simple actuator disk theory (Burton et al., 2011, ch. 3.2) is also included, as shown in Figure 3(a) (note in particular the deviation from the cubic curve before rated wind speed). The adjustments to the IEA 15-MW thrust and power curves to represent peak shaving are also mapped over to the downscaled version used to represent the 11-MW turbines. The 13-MW turbine already has a rounded corner in the thrust and power curves and, as such, is left as is.





## 2.2 Wind turbine layout

Estimating the turbine layout for projects in the U.S. pipeline that have not yet finalized their project design poses significant challenges (Mulas Hernando et al., 2023). Recognizing these challenges, we have developed a replicable framework for generating turbine layouts for U.S. lease areas. This framework adapts to the project's permitting stage or construction proximity, incorporating the most recent market information available at the time of application.

The proposed framework adopts a three-step approach. The first step involves determining the project capacity for each wind 165 lease area. When publicly available data, such as offtake agreements or announced developer plans, are accessible, they are directly utilized. Otherwise, the project capacity of each wind lease area is determined by multiplying the lease area by the estimated CD, which is the weighted-average CD specific to the state in which the wind lease area is located (Mulas Hernando et al., 2023). The second step focuses on determining the turbine rating for each wind lease area, relying again on publicly available information, such as turbine supplier agreements. In the absence of such data, the turbine rating is estimated based on 170 whether the commercial operation date (COD) for each lease is expected before or after 2026. If before 2026, the turbine rating is set at 13 MW; otherwise, the turbine rating would be 15 MW. The assumed 13- and 15-MW turbine ratings are selected based on market research from NREL's Offshore Wind Market Report (Musial et al., 2023). The final step involves generating the turbine layout. For each lease area, if a proposed turbine layout is publicly available, it is adopted with modifications to accommodate as many turbine positions as necessary, ensuring that the number of turbine positions aligns with the selected 175 project capacity and turbine rating. In cases where no proposed layout is available, the number of wind turbines is calculated by dividing the project capacity by the turbine rating. The turbines are then distributed across the lease area with uniform spacing. Note that, in our framework, equidistant layouts are used only when detailed, project-specific layout information is not publicly available—typically for lease areas that are still in the early stages of development, where no layout information is disclosed in permitting documents (e.g., COPs or official BOEM GIS data). Detailed flowcharts outlining this three-step 180 approach are provided in Figure B1, Figure B2, and Figure B3, respectively, and more discussion on generating the turbine layout is provided in Appendix B.

Using this approach, we generated the proposed layout for all 26 wind lease areas along the U.S. East Coast. We note that the layouts presented in this study are based on data collected up to March 13, 2024, and reflect project information available as of March 13, 2024. A total of 3,057 wind turbines were located within the 26 lease areas plotted, with the project capacity 185 per lease area ranging from 132 to 2,640 MW and CD ranging from 1.81 to $9.55\,\mathrm{MW\,m^{-2}}$.

## 3 Data Analysis

We analyze the results from the WRF simulations described in the previous section by defining the "wake shadow", that is, a region around a lease are that where there is an energy loss due to the wind farm cluster wakes. This approach differs from the work of others (Rosencrans et al., 2024; Pryor and Barthelmie, 2024b) that have focused on wind speed deficit when defining 190 the waked region. This section first describes the stability classification we use to analyze cluster wakes in different atmospheric conditions before providing a description of the energy loss analysis approach and definition of the "wake shadow".





## 3.1 Stability classification

Atmospheric stability is the tendency of air to resist or enhance vertical motion, which plays a critical role in weather patterns and boundary-layer processes (Stull, 1988). Several methods exist to define atmospheric stability, each based on different parameters, such as temperature gradients, turbulence, and mean wind profiles. A widely used index for assessing dynamic boundary-layer stability is the Richardson number (Ri; Kaimal and Finnigan, 1994), which is a dimensionless ratio that compares the relative contributions of buoyancy and shear forces.

To study the impact of atmospheric stability on wind turbine wakes, we choose to describe stability using the bulk Ri rather than the Obukhov length, especially given the degree of stable stratification and the size of the rotor-swept heights in an offshore environment (Rosencrans et al., 2024). In strong stability conditions, surface flows can decouple from those aloft, and therefore surface fluxes may not represent stability conditions across the rotor layer, where stability affects wakes (Quint et al., 2025). The expression for the bulk Ri is described in equation (4):

$$\mathrm{Ri} = \frac{g}{\bar{\theta}} \cdot \frac{\frac{\Delta\theta}{\Delta z}}{\left(\frac{\Delta u}{\Delta z}\right)^2 + \left(\frac{\Delta v}{\Delta z}\right)^2} \tag{4}$$

where $g$ is the gravitational acceleration, $\theta$ is the potential temperature, $\bar{\theta}$ is the mean potential temperature over a layer of atmosphere, $u$ is the horizontal wind velocity in the west-east direction, and $v$ is the horizontal wind velocity in the south-north direction. To better understand and quantify the stability difference in an offshore environment, we calculate Ri over both the near-surface layer (20 m to 50 m) and over the entire rotor layer (20 m to 300 m). Additionally, we perform this analysis across three different wind lease regions (north, central, and south; Figure 1) to assess regional variations in atmospheric stability.

Figure 4 compares the stability between the near-surface layer and rotor layer in the predefined northern, central, and southern wind lease regions. In the majority of cases (86%), the surface and rotor layers share the same stability characteristics, with all layers simulated as stable 61% of the time and unstable 25% of the time. Instances where the near-surface layer is stable but the rotor layer is unstable are extremely rare, occurring only 0.2%-1% of the time. More commonly, 13% to 14.4% of the time, the near-surface layer is unstable while the upper rotor layer remains stable. These conditions are critical for phenomena such as marine fog (Koračin et al., 2014). Overall, the difference in stability conditions across the simulation region is small, indicating minimal sensitivity in the geospatial representation of background stability. This analysis demonstrates, as also indicated in Rosencrans et al. (2024), that near-surface stability is not always representative of the deeper rotor layer. For the remainder of the paper, we will use the rotor-layer Ri for further analysis, as it is more appropriate for examining the effects of wind turbines on energy loss.

## 3.2 Wake impact identification using energy losses

In a wind farm, where multiple wind turbines are placed in clusters, the wakes of individual turbines interact with one another, leading to cumulative effects that can extend far beyond the wind farm area. These combined wakes from one or more wind farms are referred to as "cluster wakes". Often, the wake wind speed deficit map, which shows the difference in wind speed at





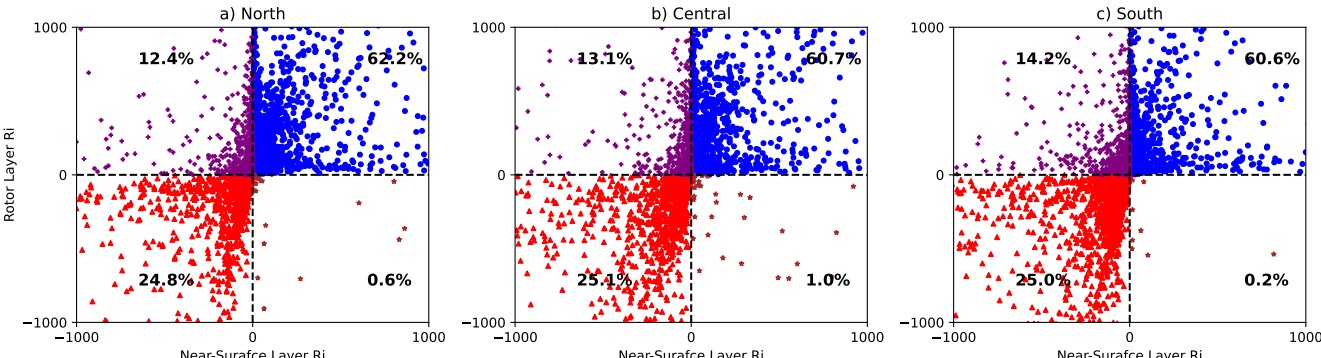

**Figure 4. Difference in Ri between the near-surface layer (20 m to 50 m) and rotor layer (20 m to 300 m) across three predefined wind lease regions.**

hub-height between the wind farm (WF) and no wind farm (NWF) simulations, is used as an indicator for assessing wind farm impacts on adjacent regions, where future wind farm development may take place. Although these maps provide important
information on the potential extent of wind farm wakes, they do not provide an accurate estimate of resulting energy loss. This disconnect arises because the velocity deficit does not directly indicate the energy loss due to the nonlinear relationship between wind speed and power.

Instead, we may obtain a better picture of the wake effects by combining the wake-induced wind speed deficit with a typical power curve (Figure 5). For instance, when the wind speed deficit ($\Delta v$) is close to or below the rated wind speed, wind turbines
in the wake region will experience significant power loss ($\Delta P_{\text{below rated}}$, represented by the blue shaded area). However, if the wake-induced wind speed (i.e., $v - \Delta v$) is greater than the rated wind speed, then no matter how large $\Delta v$ is, $\Delta P$ will be zero (shown as $\Delta P_{\text{above rated}}$, represented by the red shaded area in Figure 5). The latter situation is most likely to occur under stable stratification in an offshore environment (see Section 3.1), where flow from aloft is decoupled from the ocean surface. Therefore, a direct correlation between larger wind speed deficit and larger energy loss may not always hold true, especially in
the offshore environment.

We compute and represent wake-induced energy loss in a geospatial map using a two-step approach. First, we pass the wind speed map through the power curve to get the power map for both the NWF and WF simulations at each model output timestep (10 min). In the second step, we take the integral of the power with time and then calculate the energy loss for each location on the map with

$$\text{Energy Loss} = 100\% - \frac{\int_{t_0}^{t_1} P_{WF} \cdot dt}{\int_{t_0}^{t_1} P_{NWF} \cdot dt} \times 100\% \tag{5}$$

where $P_{WF}$ and $P_{NWF}$ are the power production from the WF and NWF simulations, respectively, and $t_0$ and $t_1$ are the time integrals used to calculate energy production. To account for variations in wind speed and atmospheric stability, we further





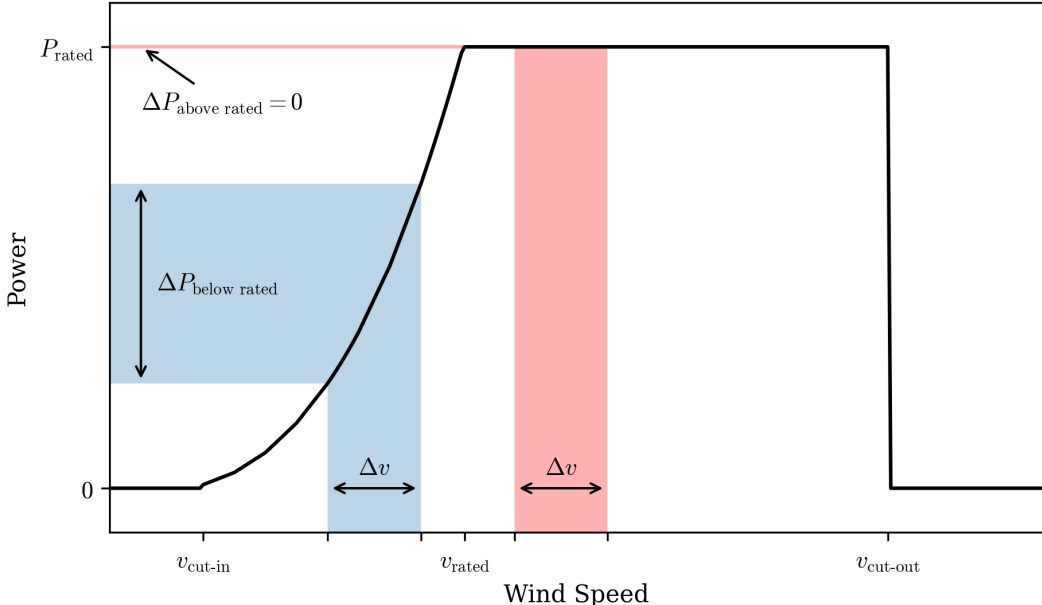

**Figure 5. A schematic diagram of the relationship between power loss ($\triangle P$) and wind speed deficit ($\triangle v$) on a typical wind turbine power curve.**

categorize the energy loss based on whether the atmosphere is stably stratified and whether the hub-height wind speed is above or below the rated wind speed ($11\,\mathrm{m\,s^{-1}}$).

To quantify the extent of wind farm wake effects, we consider 2% and 5% energy loss thresholds and calculate the total area enclosed within these thresholds on the geospatial map. These areas are referred to as the "wake shadow". We also determine the ratio, $\mathrm{F_{wakearea}}$, between the wake shadow and the total turbine area using as

$$\mathrm{F_{wakearea}} = \frac{\mathrm{Area_{wakearea}}}{\mathrm{Area_{windlease}}} \tag{6}$$

where $\mathrm{Area_{wakearea}}$ represents the wake shadow defined by a specific energy loss threshold (e.g., 2% or 5%), $\mathrm{Area_{windlease}}$ denotes
the total wind lease area within the study region. Note that the ideal threshold should ideally be informed by industry standards; however, in the absence of such guidance, we adopted the threshold values from Pryor et al. (2021a) to facilitate comparison of wind speed deficit results. A similar analysis is performed using wind speed deficit metrics to enable comparison between the two approaches. Additionally, this analysis is conducted across three different wind farm regions (north, central, and south; Figure 1) to account for the effects of cluster size and CD on the extent of the wake area.





|  | **Below-Rated** hub-height wind speed ($<11\,\mathrm{m\,s^{-1}}$); | **Above Rated** hub-height wind speed ($>11\,\mathrm{m\,s^{-1}}$); |
|---|---|---|
| **Unstable** ($Ri < 0$) | 16%, UBR | 9%,UAR |
| **Stable** ($Ri > 0$) | 45%, SBR | 30%, SAR |

**Table 3.** Frequency of occurrences during the 1-year TMY simulation ).

## 4   Results

### 4.1   Impact of atmospheric stability on wake characteristics based on energy loss versus wind speed deficit

Using the rotor-layer Ri and hub-height (150 m) wind speed from the NWF simulation, we define four distinct scenarios to compare wake characteristics based on either wind speed deficit or energy loss: 1) unstable stratification with below-rated hub-height wind speed, UBR, 2) unstable stratification with above-rated wind speed, UAR, 3) stable stratification with below-rated wind speed, SBR, and 4) stable stratification with above-rated wind speed, SAR. The most common occurrence (45%) is when the rotor layer is stable and the hub-height wind speed is below the rated power, SBR (Table 3). This 45% bin is also the most impactful condition that affects wake propagation and energy deficits. The least common combination of wind speed and atmospheric stability is unstable stratification and above-rated hub-height wind speed (UAR) with only 9% of cases.

Figure 6 shows the wind speed deficit and energy loss associated with each flow scenario. In unstable atmospheric conditions, the wind speed deficit caused by the wind farm remains nearly unchanged, regardless of whether the hub-height wind speed is below-rated (UBR; Figure 6a) or above-rated (UAR; Figure 6c). Under stable conditions, the wind speed deficit is largest when the hub-height wind speed is above-rated (SAR; Figure 6g), indicating the largest wind farm impact on wind speed. However, the story changes when wind speed deficits are translated into energy loss. Despite SAR having the largest wind speed deficit, it shows only the second-to-last impact on energy loss (Figure 6h). In fact, when wind speed is above-rated, energy loss is much smaller compared to below-rated wind speeds. The greatest energy loss occurs in SBR, where the wind speed is below-rated under stable conditions (Figure 6f). Under this scenario, power generation for wind farms like Community Wind, Attentive Energy, and Leading Light Wind could experience a greater than 30% reduction in power output. Notably, this is also the most frequent flow scenario in the region (Table 3). In summary, the analysis emphasizes a critical finding: Large wind speed deficits do not necessarily lead to large energy losses, as also shown in Lundquist et al. (2019) Figure 5c. Significant power deficits are more often associated with below-rated wind speeds, regardless of atmospheric stability (Lundquist et al., 2019).

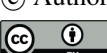

**Figure 6.** Spatial map of the average wind speed (WS) deficit and energy loss under different WS and stability scenarios from the 1-year TMY simulation. A cutoff value of -0.5 m.s$^{-1}$ for the WS deficit and 2% for energy losses are used.

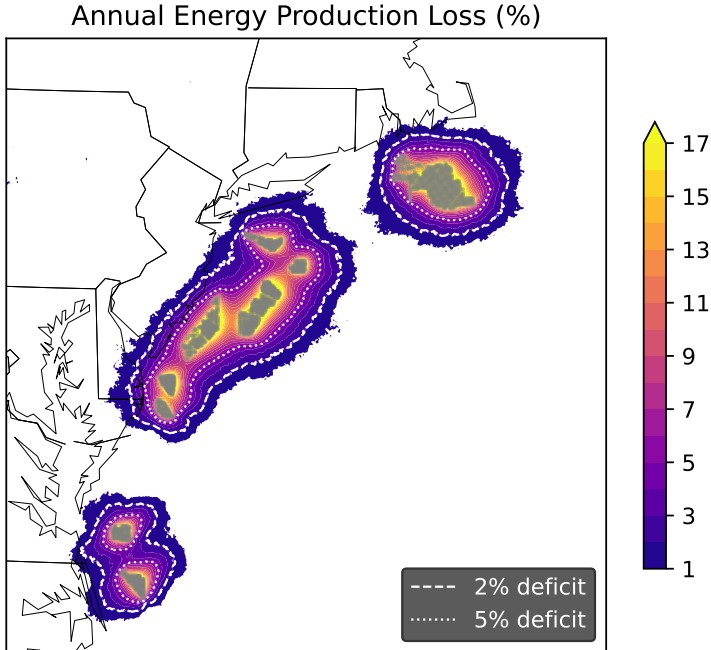

**Figure 7. Spatial map of the annual energy production loss from the 1-year TMY simulation. The dashed lines represent the 2% and 5% energy loss threshold contours.**

Figure7 presents the spatial distribution of the annual mean energy loss, the aggregate of the four scenarios in Figure 6, from the TMY simulation. Regions experiencing an annual energy loss of at least 5% are all situated near the wind lease areas. As the threshold lowers to 2%, the affected regions expand further. To further understand the regional difference in the wake extension over these offshore wind farms, Table 4 shows a comparison of wake areas over the three predefined wind farm regions (Fig.1) using 2% and 5% thresholds for energy loss and wind speed deficit. Comparing the northern and southern regions, even though the lease area in the northern region is almost four times the size of the southern region, the percentage of the wake area due to energy loss is significantly smaller in the north. Note that the CD in the northern region is half that in the southern region, which explicitly suggests that the extent of the wake area is more strongly associated with the CD rather than the size of the lease area. When the CD is around 3.23 MW/km$^2$ (e.g., northern region), the wake area associated with a 5% energy loss is approximately 2.5 times the lease area. However, in regions where the CD doubles (e.g., central and southern regions), the wake area also doubles. Lowering the energy loss threshold from 5% to 2% results in an even greater wake expansion: In high-CD regions, the wake area grows to about eight to ten times the lease area, whereas in the north, it is only four times. This result highlights the crucial role of CD in determining wake area size and underscores the need to establish an industry-acceptable energy loss threshold to maximize the area available for offshore wind development.





|  | **All** | **Northern** | **Central** | **Southern** |
|  | **Regions** | **Region** | **Region** | **Region** |
|---|---|---|---|---|
| Total lease area (km$^2$) | 9,062 | 3,673 | 4,424 | 965 |
| CD (MW/km$^2$) | 4.98 | 3.23 | 6.59 | 6.44 |
| $F_{wakearea} \geq 5\%$ energy loss | 3.68 | 2.42 | 4.70 | 4.07 |
| $F_{wakearea} \geq 5\%$ wind speed deficit | 2.82 | 1.89 | 3.63 | 2.67 |
| $F_{wakearea} \geq 2\%$ energy loss | 7.12 | 4.53 | 8.57 | 11.3 |
| $F_{wakearea} \geq 2\%$ wind speed deficit | 6.20 | 3.88 | 7.68 | 9.34 |

**Table 4.** Comparison of wind lease areas, CDs, and wake area percentages across different regions. The wake area factor $F_{wakearea}$ is defined in Equation (6).

## 4.2 Differences in wake shadow based on energy loss and wind speed deficit approach

As emphasized in the previous section, wind speed deficits do not always translate directly to energy losses. Here, we continue to express the waked area as a percentage reduction in energy or wind speed to allow a direct comparison. The distinction between using a percentage wind speed deficit as opposed to an absolute wind speed deficit is described in Appendix C. We now highlight the sensitivity of the wake shadow extension when using the energy loss approach versus the wind speed deficit approach.

Accounting all offshore wind farm lease areas, the annual mean wake area quantified using the energy deficit approach is approximately 14% to 30% larger than that defined using the wind speed deficit approach (Table 4). This greater wake expansion from the energy deficit approach is also consistently identified across all three regional wind lease areas. Figure 8 demonstrates the spatial extend of the wake area under 2% and 5% deficit scenarios for each wind lease region. Depending on the specific lease region, the wake area can be 11% to 52% larger when employing the energy loss approach compared with the wind speed deficit approach. For both thresholds, the wake area from the energy deficit approach exceeds that from the wind speed deficit approach, and the difference becomes more pronounced when the threshold increases from 2% to 5%. For instance, the wake area over southern region (Fig. 8e and 8f) defined using the energy loss approach is 21% larger than that obtained using the wind speed deficit approach when employing a 2% threshold while at a 5% threshold the difference increases to 52%. Overall, these results highlight the sensitivity of defining the "wake shadow" using the energy loss method, in contrast to the traditional wind speed deficit approach. The findings underscore that the area available for offshore wind development strongly depends on the criteria used to define the wake shadow and the industry-acceptable threshold for energy loss.





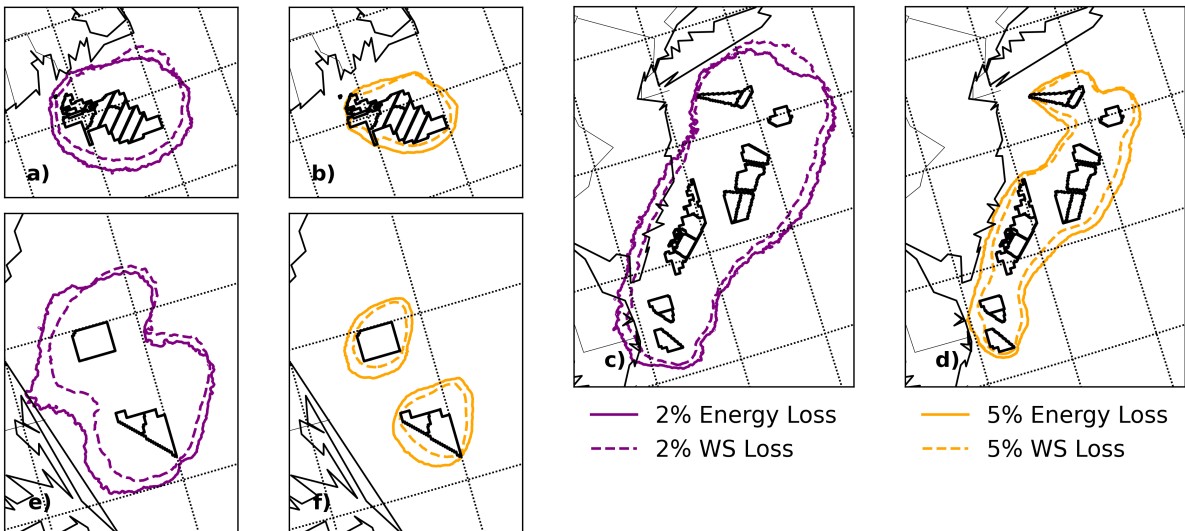

**Figure 8.** Comparison of the spatial extent of the wake area using the energy loss and wind speed (WS) deficit approaches. The purple line represents the 2% threshold, whereas the orange line represents the 5% threshold. Panels (a) and (b) illustrate the wind lease areas in the northern region, panels (c) and (d) depict those in the central region, and panels (e) and (f) show the wind lease areas in the southern region.

## 4.3 Comparison to existing literature

Lastly, we compare key findings from this study—such as the number of wind turbines, turbine ratings, CD, and wake area factors—with recently published studies (Rosencrans et al., 2024; Pryor et al., 2021a) that employ different turbine layouts and simulation methodologies. The results are summarized in Table 5.

The present study reports the highest total project capacity at 45,102 MW, substantially surpassing the estimates of Rosencrans et al. (2024) and Pryor et al. (2021a), which are 17,016 MW and 28,830 MW, respectively. This highest capacity is
315 primarily driven by the largest wind lease area and the greatest number of wind turbines, whereas the values reported by Rosencrans et al. (2024) and Pryor et al. (2021a) remain relatively comparable. Consequently, the CD is highest in this study (4.92 MW/km$^2$), followed by Pryor et al. (2021a) (3.23 MW/km$^2$) and Rosencrans et al. (2024) (2.7 MW/km$^2$). Additionally, while both Pryor et al. (2021a) and Rosencrans et al. (2024) assume a single wind turbine rating of 15 MW and 12 MW, respectively, this study incorporates a mix of 11-MW, 13-MW, and 15-MW turbines, offering a more diversified turbine
configuration.

Reducing the wind speed deficit from 5% to 2% results in a doubling of the wake area factor across all three studies, indicating a significant expansion of the wake region. Comparing our results with those of Rosencrans et al. (2024), we observe that an increase in installed CD is accompanied by a higher wake area factor, aligning with our previous conclusions (Table 4). However, despite having a lower CD, the wake area factor reported by Pryor et al. (2021a) is higher than in our study under





|  | *This work* | *Rosencrans et al. (2024)* | *Pryor et al. (2021a)* |
|---|---|---|---|
| Total Project Capacity (MW) | 45,102 | 17,016 | 28,830 |
| All Lease Area (km$^2$) | 9,062 | 6,194 | 6,566 |
| Number of Wind Turbines | 3,057 | 1,418 | 1,922 |
| Average Installed CD (MW/km$^2$) | 4.98 | 2.75[a] | 4.39 |
| WTR (MW) | 11, 13, 15 | 12 | 15 |
| F$_{\text{wakearea}} \geq 5\%$ Wind Speed Deficit | 2.82 | 1.71 *(1.80)* | 3.73 |
| F$_{\text{wakearea}} \geq 2\%$ Wind Speed Deficit | 6.20 | 4.02 *(3.90)* | 7.69 |

[a]Rosencrans et al. (2024) identified their average installed CD as 3.14 MW/km$^2$ focusing on areas within the lease boundaries. When considering the irregular arrays and spacing between their lease areas, we find an average installed CD of 2.75 MW/km$^2$. The italic values are calculated using simulation from Rosencrans et al. (2024) that has 0% added TKE.

**Table 5.** Comparison of key findings across three different studies: For Pryor et al. (2021a), the wake area factor is calculated as the average factor of the 11-flow scenario from Table 3 of their paper.

both wind speed deficit scenarios. This discrepancy can primarily be attributed to two key factors. First, the difference in the equations used to calculate wind speed deficit plays a crucial role. Equations (7) and (8) are the equation used in our study and in Pryor et al. (2021a), respectively, for calculating wind speed deficit.

$$\text{Wind Speed Deficit} = \left( \frac{\frac{1}{n}\sum_{i=1}^{n} WS_{WF} - \frac{1}{n}\sum_{i=1}^{n} WS_{NWF}}{\frac{1}{n}\sum_{i=1}^{n} WS_{NWF}} \right) \times 100\% \tag{7}$$

$$\text{Wind Speed Deficit} = \frac{1}{n}\sum_{i=1}^{n} \left( \frac{WS_{WF} - WS_{NWF}}{WS_{NWF}} \right) \times 100\% \tag{8}$$

where $n$ represents the total number of hourly output from the year-long simulation, $WS_{WF}$ and $WS_{NWF}$ are the hub-height wind speed from the WF and NWF simulations. The primary distinction between these methodologies lies in the normalization of wind speed. In our study, normalization is conducted at the final stage using the mean wind speeds from the WF and NWF simulations (Eq. (7)). Conversely, in Pryor et al. (2021a), normalization occurs at each output timestep before being averaged over time (Eq. (8)). Note that we calculated wind speed deficit using equation (7) because it has the same mathematical form

as equation (5), which is the standard method for estimating annual power loss. This is to ensure consistency and comparable assessment in our analysis by applying the same formulation to both wind speed deficit and energy loss.

The impact of these differing approaches is illustrated in Figure 9 from our simulation and in Figure C2 from Rosencrans et al. (2024) (see Appendix C). Notably, the wake area is considerably larger when using Equation (7) compared to Equation (8). For instance, when considering a 2% wind speed deficit, the wake area calculated with Equation (7) is twice as large as

that obtained using Equation (8). This finding underscores the sensitivity of wake area estimation to the choice of wind speed deficit calculation methods. Future research should account for these methodological differences and exercise caution when





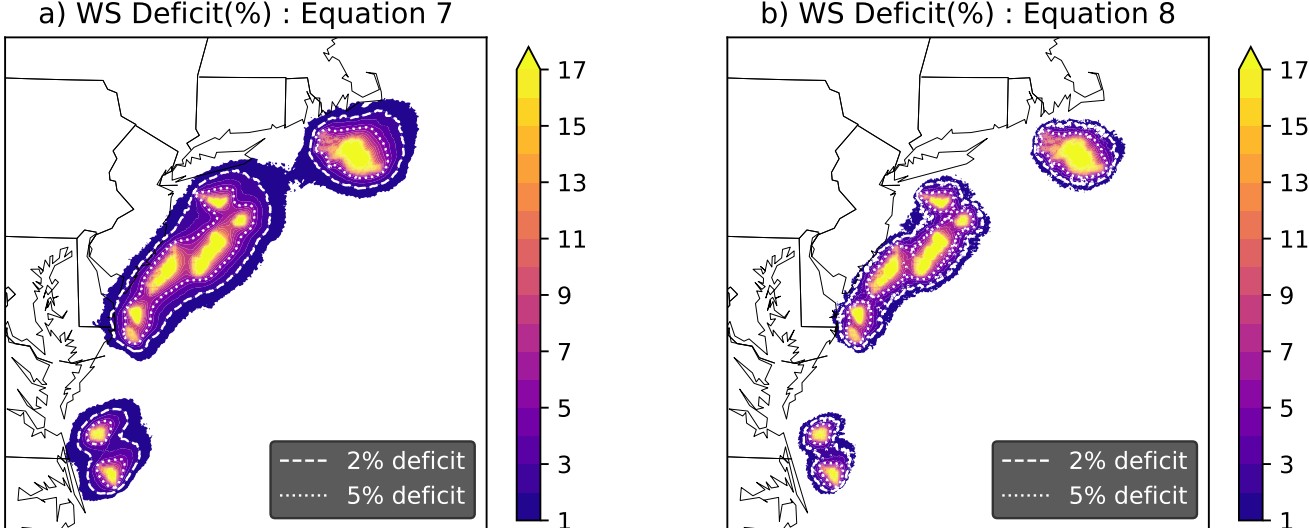

**Figure 9. Comparison of annual wind speed deficit calculated using Equation** (7) **and** (8)) **from the one-year TMY simulation.**

selecting an appropriate approach. Ultimately, wind speed measurements from the far-wake offshore region are necessary to determine which normalization method is most suitable for wind speed deficit calculations.

The second reason of discrepancy in the results is primarily related to differences in simulation methodologies. Both our study and that of Rosencrans et al. (2024) employ year-long simulations to represent the regional wind climate., whereas Pryor et al. (2021a) conduct 11 five-day representative wind flow scenarios. This methodological difference likely introduces variations in wind statistics, which could significantly influence the wake area factor calculations.

The impact of added TKE on the wake area is also examined using simulation from Rosencrans et al. (2024). Reducing TKE from 100% to 0%, the $F_{\text{wakearea}}$ increases from 1.71 to 1.80 under 5% wind speed deficit scenario but decreases from 4.02 to 3.90 under 2% wind speed deficit scenario (Fig.C2). Overall, these changes are insignificant, consistent with recent finding indicating that meteorological impacts of added TKE is the least sensitive at the hub-height level (Quint et al., 2024).

Overall, these findings highlight the complex interplay of multiple factors, including turbine design, simulation techniques, and analytical methods, in determining the final wake area. This complexity underscores the need for further research to better understand and standardize wake impact assessments for future offshore wind development.

## 5 Conclusions

This study presents new results for cluster wake-induced energy losses in the U.S. East Coast offshore wind lease areas. In contrast to previously reported work, we focus on energy loss rather than wind speed deficit when defining the wake-affected region. We have also developed a replicable framework for generating realistic turbine layouts based on publicly available



information, focusing on U.S. offshore wind lease areas and adapted the TMY approach to better represent regional wind
climate in the numerical simulations used to evaluate wakes.

The study identifies a significant difference between the resulting wake extents when defining the "wake shadow" using
the traditional wind speed deficit approach and and when using the energy loss method. Under a 5% threshold, the wake area
associated with the energy loss approach can be 30% larger in size compared to that defined using the wind speed deficit ap-
proach. These results indicate that the wind speed deficit approach may underestimate the wake area. The findings demonstrate
considerable sensitivity to the chosen method for defining wake areas and emphasize the importance of incorporating both the
wind speed deficit and energy loss approaches in offshore wind development assessments.

Our findings also underscore the notable difference in stability conditions when comparing the near-surface layer to the
entire rotor layer in offshore environments. This disparity suggests that greater caution should be exercised when selecting
the appropriate stability definition for analyzing wind farm impacts. Consistent with previous research, stable atmospheric
conditions lead to larger wind speed deficits, particularly when hub-height wind speeds exceed the rated value. However, a
key insight from this study is that large wind speed deficits do not necessarily translate into significant energy losses. In fact,
the greatest energy loss occurs when wind speeds are below-rated with stable stratification. This result highlights the need to
consider both wind speed deficits and energy losses when evaluating the wake effects of offshore wind farms.

Additionally, the results indicate that the extent of the wake area is strongly influenced by the capacity density (CD) and the
selected cutoff threshold. A wind lease region with larger CD generally suggests a larger wake area. Lowering the energy loss
threshold from 5% to 2% results in significant wake expansion, with the wake area potentially increasing to seven to ten times
the size of the lease area. This finding highlights the crucial role of CD in determining the wake area and underscores the need
to establish an industry-acceptable energy loss threshold to maximize the area available for offshore wind development.

A comparison with recently published studies on offshore wind wake assessments demonstrates the complex interplay of
multiple factors in determining the final wake area, including the (1) turbine design and layout, (2) simulation techniques,
and (3) analysis methods and metrics employed. The variability in the results arising from these factors highlights the need
for further research to standardize wake assessment methodologies and develop more robust, universally applicable wake
characterization techniques. Future studies should aim to bridge the gaps between various assessment methods, ultimately
leading to more accurate and consistent wake impact predictions across different wind farm configurations and environmental
conditions.

*Data availability.* The data and files that support this work are publicly available. The ERA5 forcing data can be downloaded from the
ECMWF Climate Data Store at https://doi.org/10.24381/cds.bd0915c6 (Hersbach et al., 2020). Shapefiles including the bounding extents
of the lease and call areas are available at https://hub.arcgis.com/datasets/709831444a234968966667d84bcc0357/explore. (BOEM,
2021). Individual turbine coordinates, power and thrust curves and WRF namelists for NWF and WF simulations can be obtained at https:
390   //doi.org/10.5281/zenodo.15078171.The simulation output data will be available in HDF5 format upon request.



## Appendix A: Validation of GPU-Accelerated WRF Model AceCAST Against the Standard CPU-Based Model

In October 2023, Veer Renewables published a validation report summarizing AceCAST—a GPU-accelerated version of the WRF model developed by TempoQuest—and the standard CPU-based WRF model (Veer, 2023). A full-year TMY was simulated over a U.S. offshore Atlantic domain very similar to that simulated in this study. However, a coarser inner 3-km domain
nested within a 9-km domain was employed. Wind farms spanning from Delaware to Massachusetts were modeled, using confidential client-provided wind turbine characteristics and locations. Comparisons were performed between WRF 4.4.2 and an equivalent AceCAST build, both based on the same physics options described in Table 1.

Figure A1 compares the mean modeled wind speeds, power, and TKE over the TMY between the standard WRF model and AceCAST. As shown in the figure, the agreement between the two models is excellent, with differences in wind speed and
power deviations less than 1.5% and TKE deviations less than 2%, according to:

$$perf_{diff} = 1 - \frac{V_a}{V_w} \tag{A1}$$

where $V_a$ is the AceCAST variable (either wind speed, power, or TKE) and $V_w$ is the corresponding WRF variable.

These differences are of similar magnitude to those observed when the WRF model is launched on different high-performance computing clusters (Hahmann et al., 2019). When comparing the mean net capacity factor across the different lease areas, we
observe deviations between the WRF model and AceCAST ranging from -0.22% to 0.03%, with an average of -0.12%.

Next, we perform timeseries analyses of the AceCAST and WRF simulations. In Figure A2, we provide a snapshot of key atmospheric variables for January 2019. As shown in the figure, the agreement between the WRF and AceCAST timeseries is excellent, with only small deviations in atmospheric parameters. Across the four variables shown in A2, correlations between the WRF and AceCAST simulations exceeded 0.99, and the distributions were nearly identical.

The results from this validation study demonstrate that AceCAST is a suitable alternative to the standard CPU-based WRF model. Whether looking at mean atmospheric parameters modeled across the entire domain or timeseries analysis at specific coordinates, we find strong agreement between AceCAST and the WRF model.



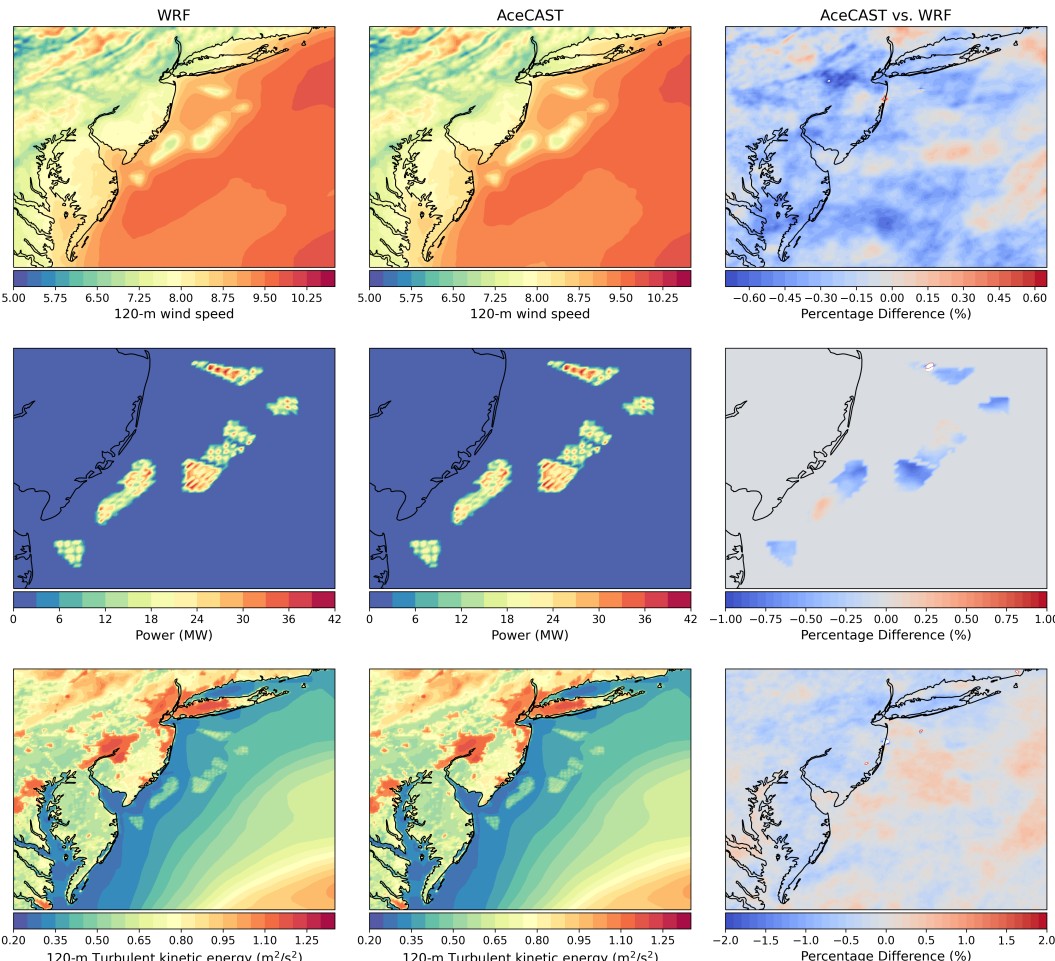

**Figure A1.** Mean maps of 120-m wind speed (top row), power (middle row), and 120-m TKE (bottom row), modeled using the WRF model (left column) and AceCAST (center column). Percentage differences between the WRF and AceCAST results are shown in the right column.

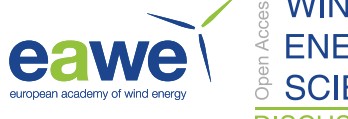

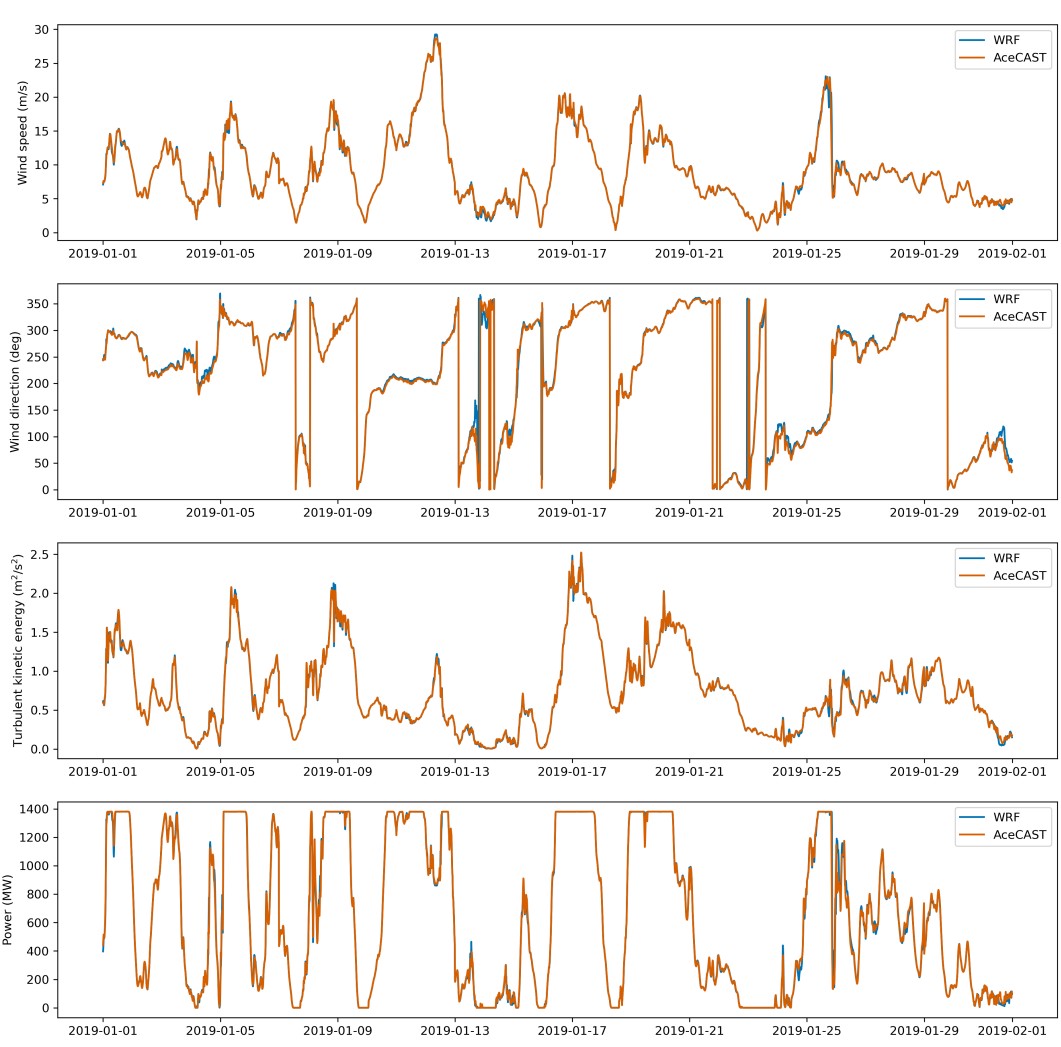

**Figure A2. Timeseries visualization of key atmospheric parameters for January 2019 within a selected lease area.**





## Appendix B: Details of Wind Turbine Layout Generation Approach

Following section 2.2, here we provide further detail on the three-step approach of generating turbine layout using our proposed
framework. The first step is to estimates the PC of each wind lease area (Fig. B1) using either public available data (e.g., offtake
agreement) or multiplying the lease area by the estimated CD, which is the weighted-average CD specific to the state[1] in which
the wind lease area is located (Mulas Hernando et al., 2023). Estimating PC for each wind lease area is the first and most critical
step of our proposed turbine layout generation approach because it ensures the layout is realistic, aligned with regulatory and
financial expectations, and feasible within the constraints of the lease area. The second step focuses on assigning turbine rating
for each wind lease area (Fig. B2). For projects lacking turbine supplier agreements and with a COD before 2026, we assume
the use of 13-MW turbines. For projects expected to reach the COD in 2026 or later, we use 15-MW turbines, reflecting current
industry trends and supplier announcements [2].Finally, the third step involves defining a turbine layout (Fig. B3). If a proposed
layout exists in the public, it is adopted or adjusted as needed to match the expected PC. For instance, Revolution Wind (OCS-
A 0486) is a 704 MW capacity offshore wind farm under construction off the coast of Rhode Island. The actual layout and
turbine rating are already available to public and thus, we adopted their turbine positions as the layout. In the case of Bay State
Wind (OCS-A 0500), located offshore Massachusetts and currently in the permitting stage, the proposed layout by BOEM is
modified by removing the 30 turbine positions closest to the lease area boundary in order for the total PC to be aligned with the
estimated PC. For leases without proposed layouts—typically those in early development stages with no permitting documents
available (half of the offshore wind projects examined in this study)—the number of turbines is calculated by dividing the
project capacity by the turbine rating. Turbines are then distributed uniformly across the lease area using equidistant spacing,
which serves as a default layout assumption. Note that, in our framework, equidistant layouts are used only when detailed,
project-specific layout information is not publicly available. In contrast, when turbine layout data is available—along with
other market-relevant data such as project capacity from offtake agreements, turbine ratings, or regionally-informed capacity
density assumptions—our framework integrates that information to produce more representative and realistic layouts.

The layouts presented in this study are based on data collected up to March 13, 2024, and reflect the project information
available as of March 13, 2024. Details of the layouts for individual lease areas after applying the layout generation approach
are given in Table B1. Figures B4)a and B4)b show the spatial pattern of CD and WTR across the wind lease areas in the
study region. Notably, CD tends to increase as one moves southward, where newer wind lease areas are located. This trend
can be attributed to two primary factors: First, the spacing between turbines in new wind lease areas becomes more compact
(transitioning from the uniform wide spacing of the MA/RI lease areas to a combination of wide and narrow corridors); second,

---

[1]Based on weighted-average capacity densities by state in Mulas Hernando et al. (2023), we use 3 MW/km$^2$ for MA and RI, 4 for NJ, 6 for MD and VA,
and 7 for NC, NY, and DE lease areas. These are the states to consider according to BOEM state leasing activities (BOEM, 2021).

[2]South Fork Wind, Revolution Wind, and Sunrise Wind are expected to use 11-MW turbines based on turbine supplier agreements, whereas Vineyard
Wind is installing 13-MW turbines (McCoy et al., 2024). Based on this, we assume that projects with a COD before 2026 and no supplier agreements will use
13-MW turbines. Projects like Empire Wind and CVOW-C are projected to use 14-MW to 15-MW turbines based on supplier agreements. Vestas is focusing
on the 15-MW V236 model, GE is prioritizing the 15.5-MW Haliade-X, and, although Siemens Gamesa has not fully revealed its strategy for new turbine
models, it has secured a supply agreement for 14.7-MW turbines in the United States (McCoy et al., 2024). This assessment assumes a trend toward 15-MW
turbines for mid-to-long-term projects post-2025, but we recognize that there may be potential for further turbine upscaling.




larger and more powerful wind turbines are being proposed for these new wind areas. Using the proposed framework for layout generation, the WTR for each lease area is also determined, with three turbine capacities—11 MW, 13 MW, and 15 MW—being applied. Most lease areas are assigned 15-MW turbines, as their CODs are yet to be determined. However, for areas such as South Fork Wind, Revolution Wind, and Sunrise Wind, the turbine ratings have already been established based on agreements
with their turbine suppliers. Figure B4c provides a detailed view of the exact wind turbine locations/layout within the MA/RI lease areas, with a prescribed spacing of 1 nautical mile between each turbine. For Revolution Wind (OCS-A 0486) and South Fork Wind (OCS-A 0517), the 11-MW wind farm layouts were predetermined based on publicly available information. In contrast, the layouts for the remaining wind lease areas were established using the proposed framework.

Even though this framework is specific applied to the U.S. offshore wind lease area, we believe it is, in fact, applicable to
450 wind farms globally—in the sense that it offers a replicable approach for generating realistic turbine layouts using publicly available information. Leveraging data such as offtake agreements and permitting documents to inform layout assumptions is a practice that can be applied in any region where such data exists.

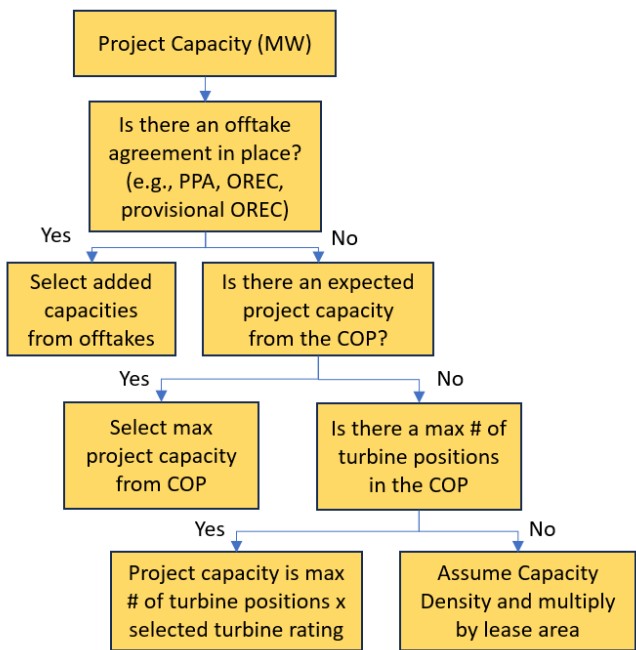

**Figure B1. Flowchart outlining the process of determining the project capacity for each wind lease area.**



| Lease Number | Project Capacity (MW) | WTR (MW) | Number of Turbines | CD (MW/km$^2$) |
|---|---|---|---|---|
| OCS-A 0482 | 1,980 | 15 | 132 | 6.97 |
| OCS-A 0483 | 2,640 | 15 | 176 | 5.87 |
| OCS-A 0486 | 704 | 11 | 64 | 1.81 |
| OCS-A 0487 | 924 | 11 | 84 | 2.08 |
| OCS-A 0490 | 1,815 | 15 | 121 | 9.55 |
| OCS-A 0498 | 1,095 | 15 | 73 | 3.58 |
| OCS-A 0499 | 2,700 | 15 | 180 | 6.53 |
| OCS-A 0500 | 2,280 | 15 | 152 | 3.0 |
| OCS-A 0501 | 806 | 13 | 62 | 3.05 |
| OCS-A 0508 | 2,580 | 15 | 172 | 7.63 |
| OCS-A 0512 | 2,070 | 15 | 138 | 6.45 |
| OCS-A 0517 | 132 | 11 | 12 | 2.4 |
| OCS-A 0519 | 975 | 15 | 65 | 9.11 |
| OCS-A 0520 | 2,325 | 15 | 155 | 4.46 |
| OCS-A 0521 | 2,235 | 15 | 149 | 4.43 |
| OCS-A 0522 | 1,605 | 15 | 107 | 3.0 |
| OCS-A 0532 | 1,155 | 15 | 77 | 3.36 |
| OCS-A 0534 | 2,040 | 15 | 136 | 4.96 |
| OCS-A 0537 | 2,025 | 15 | 135 | 7.0 |
| OCS-A 0538 | 2,745 | 15 | 183 | 8.04 |
| OCS-A 0539 | 3,000 | 15 | 200 | 5.88 |
| OCS-A 0541 | 1,290 | 15 | 86 | 4.01 |
| OCS-A 0542 | 2,400 | 15 | 160 | 7.05 |
| OCS-A 0544 | 1,320 | 15 | 88 | 7.59 |
| OCS-A 0549 | 1,320 | 15 | 88 | 4.02 |
| OCS-A 0559 | 930 | 15 | 62 | 5.89 |

**Table B1.** Summary of the project capacity, WTR, number of turbines, and CD for all 26 lease areas examined in this study.



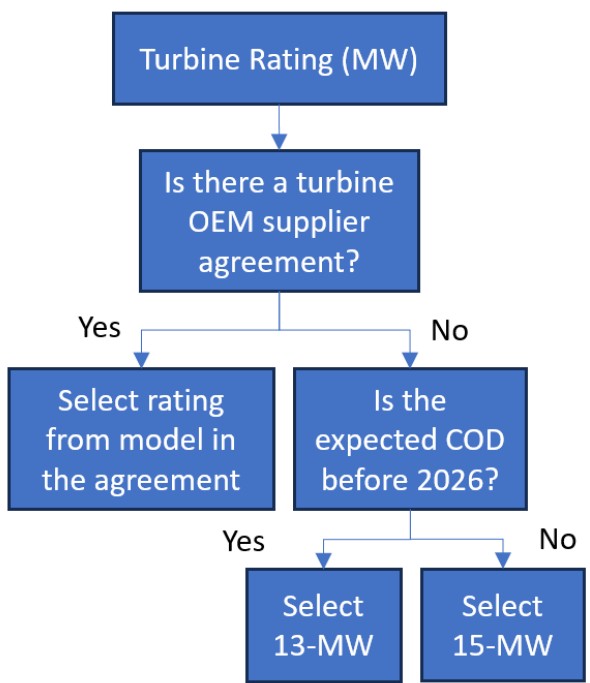

**Figure B2. Flowchart outlining the process of determining the turbine rating power.**

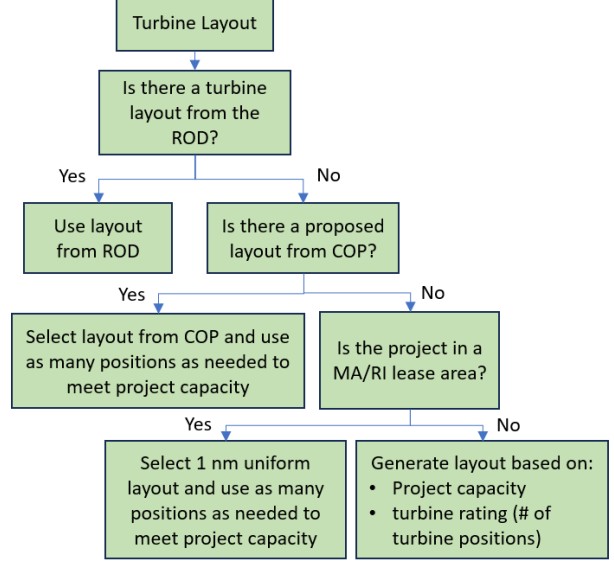

**Figure B3. Flowchart outlining the process of determining the turbine layout.**





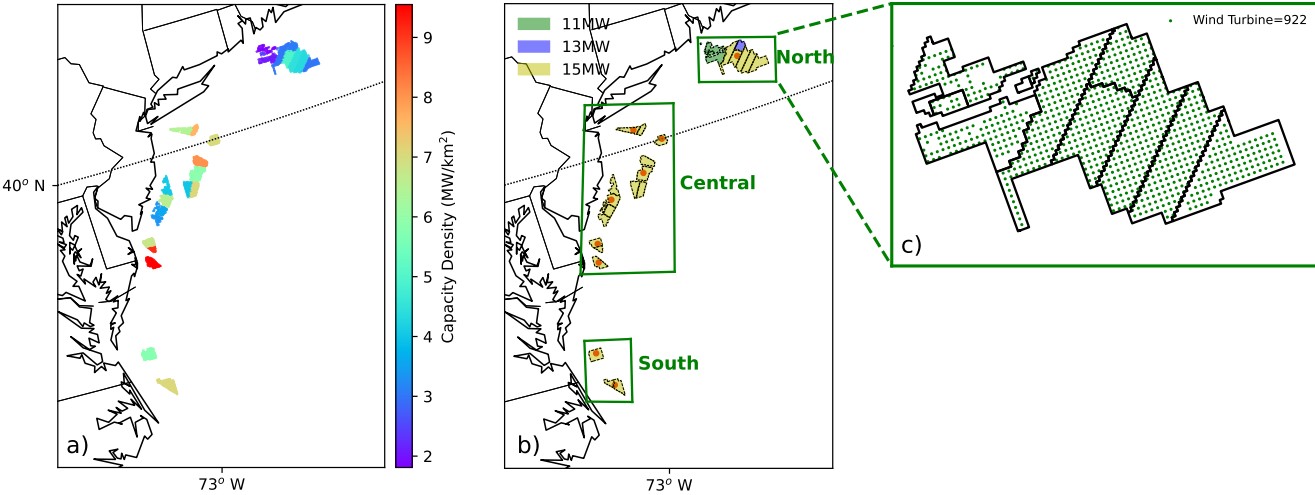

**Figure B4.** Spatial pattern of a) CD and b) WTR for all the examined lease areas along the U.S. East Coast; the orange dots are locations where data are extracted to calculate the atmospheric stability. c) An enlarged view of the wind turbine locations/layout over the MA/RI lease areas.

**Appendix C: Characterization of wind speed deficits**

While we have focused on defining wind speed loss as a percentage to facilitate a more direct comparison to energy loss, other
studies have used absolute wind speed deficits (expressed in SI units, m/s) to characterize the wake area. Figure C1 illustrates the extent of the wake based on the wind speed deficit, expressed as an aboluste value (m/s) and as a percentage (%). In this analysis, contour thresholds of -0.2 and -0.5 m/s were applied for the SI units, whereas the corresponding thresholds of 2% and 5% were used for the percentage-based representation. The difference in the total wake area defined by the -0.2 m/s and 2% contours is smaller than 1%, whereas the areas defined by the -0.5 m/s and 5% contours are almost identical. However,
this does not imply a linear relationship between the wake area defined by the wind speed deficit expressed in SI units and as a percentage. In fact, a previous study (Rosencrans et al., 2024) has shown that such a relationship does not hold, particularly when the wind speed deficit increases ($<$ -1.5 m/s). The observed correspondence appears constrained to the threshold values examined in this study. Nevertheless, this correspondence is important, as it enables direct comparison of wake areas quantified using both wind speed deficit and energy loss expressed in equivalent percentages.

Regarding the distinction between methodologies lies in the normalization of wind speed, Figure C2 further illustrate using data from Rosencrans et al. (2024). Consistent with the findings in Section 4.3, the wake area estimated using Equation (7) is approximately twice as large as that obtained using Equation (8). This highlights the sensitivity of wake area estimation to the choice of wind speed deficit calculation method.



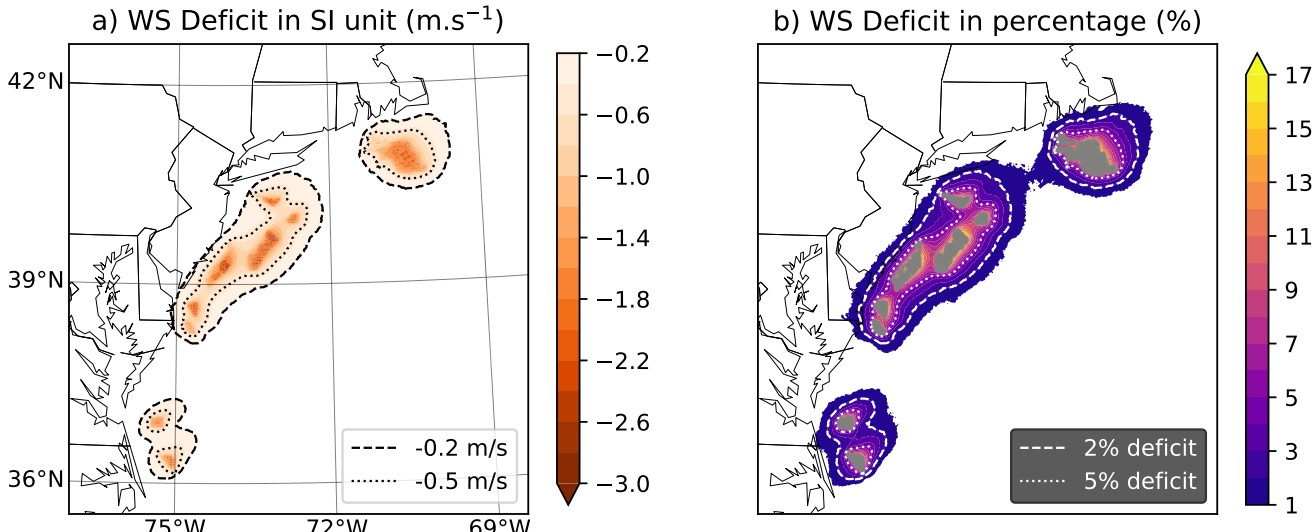

**Figure C1.** Spatial map of annual wind speed (WS) deficit expressed in SI units $(\mathrm{m.s^{-1}})$ and as a percentage (%) from the 1-year TMY simulation.

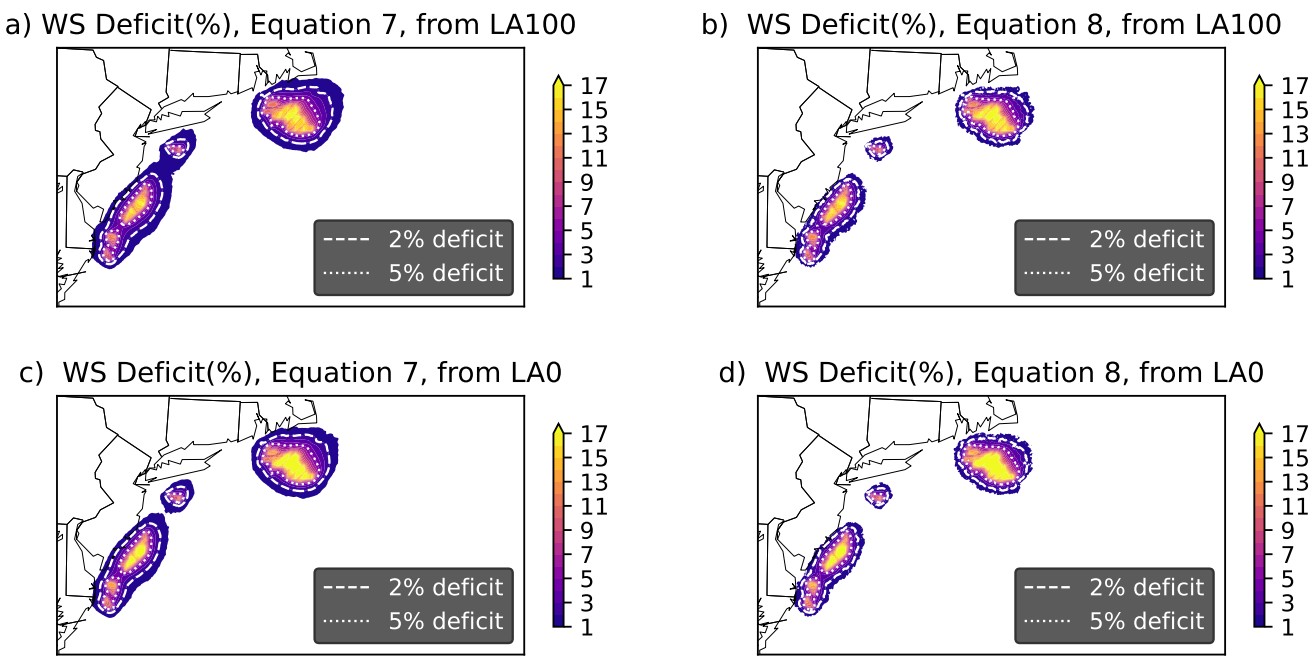

**Figure C2.** Similar as Figure 9 but using data from **Rosencrans et al. (2024)**; LA100 and LA0 represent year-long simulation with 100% and 0% added TKE.





*Author contributions.* All coauthors played an important role in this paper. Following the CRediT taxonomy, each coauthor contributed to
470 the following: GX contributed to the conceptualization, methodology, formal analysis, investigation, data curation, writing - original draft
& editing. MO contributed to the conceptualization, software, visualization, and writing - original draft & editing. GD contributed to the
supervision, project administration, funding acquisition, conceptualization, methodology, investigation and writing - original draft & editing.
MS contributed to the conceptualization, visualization, methodology, writing - review & editing. DMH contributed to the conceptualization,
methodology, writing – original draft & editing. JKL contributed to the conceptualization, investigation and writing - review & editing. AK
methodology, contributed to the visualization, investigation and writing – original draft & edits. MSG contributed to the methodology and
writing - review & edits. PM contributed to writing - review & edits. WM contributed to the conceptualization, funding acquisition and
writing - review & edits.

*Competing interests.* At least one of the (co-)authors is a member of the editorial board of *Wind Energy Science*. Furthermore, Mike Optis
is the founder and president of Veer Renewables, a for-profit consulting company that uses a wind modeling product, WakeMap, which is
480 based on a similar numerical weather prediction modeling framework as the mesoscale simulations described in this paper.

*Acknowledgements.* This work was authored in part by the National Renewable Energy Laboratory, operated by Alliance for Sustainable
Energy, LLC, for the U.S. Department of Energy (DOE) under Contract No. DE-AC36-08GO28308. Partial funding was provided by DOE's
Office of Energy Efficiency and Renewable Energy Wind Energy Technologies Office and funded in part by the Bureau of Ocean Energy
Management through an interagency agreement with DOE. Partial funding was also provided by the National Offshore Wind Research
and Development Consortium (NOWRDC) to carry out a Joint Industry Project investigating Multi-fidelity Modeling of Offshore Wind
Inter-array Wake Impacts to Inform Future U.S. Atlantic Offshore Wind Energy Area Development under CRD-23-24539-0. This material
is partially based upon JKL's work supported by the Massachusetts Clean Energy Center and the Maryland Energy Administrations as
well as the U.S. Department of Energy's Office of Energy Efficiency and Renewable Energy (EERE) under the Wind Energy Technologies
Office (WETO) Award Number DE-EE0011269. The views expressed herein do not necessarily represent the views of the U.S. Department
of Energy or the United States Government, the Maryland Energy Administration, or the Massachusetts Clean Energy Center. The U.S.
government retains certain rights in intellectual property under CRD-23-24539-0. This publication does not necessarily reflect the views of
NOWRDC or the U.S. government, and NOWRDC nor the U.S. government makes no representations or warranties and has no liability for
any of its contents. The research was performed using computational resources sponsored by DOE and located at the National Renewable
Energy Laboratory.
During the preparation of this work, the authors used ChatGPT in order to improve language and readability. After using this tool, the
authors reviewed and edited the content as needed and take full responsibility for the content of the publication.



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
