# Peer review of "Understanding Cluster Wake-Induced Energy Losses off the U.S. East Coast"

_Wind Energy Science, 2025_

## Referee Comment (RC3)

**Paper Review - WES-2024-154**

Title: Understanding Cluster Wake-Induced Energy Losses off the U.S. East Coast

September 17, 2025

In this paper, the authors use a GPU-accelerated version of WRF to investigate the impact of cluster wakes off the US East Coast. In particular, the authors consider both existing and planned wind farms using available (where possible) and derived data regarding project capacity, capacity density, wind turbine rating and layout. The authors run one year simulations, with and without wind farms, assembling the most representative months among 24 years of atmospheric conditions. The authors introduce the concept of "wake shadow", by transforming wind into power losses by means of a typical turbine power curve. Using this metric and the same percent thresholds as for wind losses, the authors show that the area affected by wind farm wakes may increase by up to 30% compared to the standard approach that solely focuses on wind speed. Additional aspects such as the effect of stability on wake shadow and inconsistencies in calculating metrics among different studies are also discussed. The main key findings of this study (claimed by the authors and identified in the manuscript) are:

1 large wind speed deficits do not necessarily translate into significant energy losses

2 power losses are most pronounced under stable conditions and below rated conditions

3 given a threshold, the area affected by wakes is highly correlated to the capacity density (this is independent of the adopted metric).

While I think this study focuses on a very relevant topic for the wind energy community, it features very little novelty in my opinion, at least in its current framing. Regarding the first of the key findings, I believe that it is common knowledge that wind turbines do not harvest all available power above rated conditions. So it is quite obvious to me that, while wakes may be strong in these cases, power deficits are zero or close to zero. On the other side, I must recognize that there is a large tendency in literature to show wind deficit maps instead of power deficit maps, so the metric in itself is justified and useful. In my opinion, also the second key finding is almost common knowledge at this point, as it is very easy to find studies (both using WRF or micro-scale models such as LES) where stronger wake deficits and persistence are observed under stable conditions. Notably, when these studies investigate the physical reasons which dictate this behavior, they are still very valid and welcomed in my view, but unfortunately this is not the case for the present manuscript. The third finding is the greatest novelty of the study in my opinion, and I am surprised it is not even mentioned in the abstract.

For the reasons above, I personally don't recommend the publication of this study in the Wind Energy Science Journal, at least in its current form and framing. In order to be eligible for publication, the authors should in my opinion expand some of the interesting thematic areas that are briefly touched throughout the paper and shift the focus towards an assessment of different metrics and guidelines for their use in addressing the magnitude of cluster wake effects, which is in my opinion novel and of interest. Specifically, the authors are encouraged to expand on the following thematic areas:

1 try to differentiate between intra-farm and cluster wakes, producing estimates regarding the fraction of each of these effect on the total loss, which is what the authors solely considered. This would probably require to run additional simulations where the north, central or south lease areas are removed (or simulations of existing vs existing + new lease areas).

2 try to expand the analysis on stability, introducing additional metrics such as the Obukhov length $L$ and/or the lapse rate inside the ABL (which gives a clear measure of stability). In fact, $Ri$ and $L$ are a proxy for the exact stability conditions throughout the entire ABL and are the only available measure when performing observations (in absence of a thermodynamic profiler, for example). In this case the authors have exact information on the stability, so why not use it.

3 try to restructure the paper narrowing focus on the comparison of different metrics to assess wake effects (thereby introducing the valuable "wake shadow" metric), and assessing their performance under different stability conditions, capacity density and lease areas. The authors should use these metrics to offer some insights on whether, using a fixed total capacity, it is better to have smaller and denser or a larger and less dense lease area. In my opinion this is not an easy question. Maybe the actual "affected area" is very similar in both cases, so the small area is convenient due to lower maintenance and installation costs. Also this point would probably require to run additional simulations using different capacity densities, making the manuscript less "realistic" but more insightful.

These three comments are expanded in Sec. 1, while specific and technical remarks are outlined in Sec. 2.

**1   General Comments**

**1.1   Comment 1**

In the abstract – as well as in the very title – the authors claim that they want to assess cluster wake effects (line 1: this study seeks to advance our understanding of energy losses caused by wind farm cluster wakes). This is further highlighted throughout the paper, but the modality used to carry out the study is not in line with the objective of understanding cluster wake effects. In fact, by looking for example at line 74, it is misleading to refer to "magnitude of the energy deficit for downstream wind farms", as the authors only consider WF (wind farms) and NWF (no wind farms) situations. Notably, energy deficits of a downstream wind farm in the form proposed by the authors are affected by both itself (the downstream wind farm) and upstream clusters. A deficit of 30%, claimed by the authors and observed inside some wind farms (line 271: under this scenario, power generation for wind farms like Community Wind, Attentive Energy, and Leading Light Wind could experience a greater than 30% reduction in power output)

is completely misleading, some of this deficit is there just because those wind farms are generating power, hence is not a power reduction of the wind farm. The research question, which the authors should try to address, is what part of this 30% reduction is due to cluster wakes effects.

This uncovers a limitation of the "wake shadow" approach, in my opinion, where velocity deficits are transformed into energy deficits. This metric measures the energy that can be extracted from any given region of the flow if a specific wind turbine is placed at this location. If one constructs the percent difference of this metric – like the authors do – by comparing the NWF and WF cases, the resulting map makes little sense in the context of addressing cluster wake effects, as it represents the energy deficit w.r.t. the case where each turbine behaves as if it operated in isolation. As a consequence, the deficit the authors are looking at contains intra- and extra-farm wakes, as well as blockage effects. To isolate the effect of neighboring clusters one should substitute the NWF case with the case where only the cluster of interest is present. Only at this point the metric provides the energy loss with respect to an ideal case where no wake effects from neighboring clusters are present. I believe that this is a crucial limitation of this study and of the "wake shadow" metric, and it should be highlighted. In an operational setting, where one wants to place a new wind farm within a new lease, the procedure to address cluster wake effects should be the following in my opinion. First, run a simulation with the new cluster only and second, include all existing neighboring wind farms. Third, use the "wake shadow" to calculate energy losses due to neighboring wind farms (cluster wake effects). NWF simulations make little sense to me in this context.

**1.2 Comment 2**

The authors use the Richardson number, which provides the ratio of buoyancy vs shear forces within the flow. While this number is able to tell if the flow is buoyancy or shear dominated, it does not provide a clear distinction between stable, unstable or neutral static stability, which is what affects the wake. A better metric to use would have been the mean lapse rate inside of the boundary layer, which provides exact information regarding the flow stability. Flows with Richardson number close to one (both positive and negative) can be either statically stable or unstable. It would have been interesting to see how $Ri$ and $L$ relate to this (and with each other, see for example Basu et al., 2014). Also, it would have been extremely interesting, downstream of the analysis proposed in Comment 1, to directly relate cluster wake losses to $Ri$, $L$ and $\gamma$ (the lapse rate) on a diagram, instead of a table, to investigate if any clear dependency is present.

**1.3 Comment 3**

While the paper aims to "advance our understanding of energy losses caused by wind farm cluster wakes", I think it does very little in this sense. I suggest the authors to restructure the paper with the objectives of

- understanding the **sole** impact of cluster wakes on new lease areas (by running these areas individually and with neighboring wind farms).

- addressing the effect of stability on these losses by using all available stability metrics ($Ri$, $L$, $\gamma$).

- relating different metrics used to compute losses (wind deficit and energy deficit using both formulas 7 and 8 in both cases) to the actual power deficit experienced by the planned wind farms in the isolated vs waked conditions. Make considerations on capacity density vs affected area with fixed total capacity.

if this framing (or similar) will be used for the revised manuscript and the analysis will be expanded, I will agree to suggest consideration of this manuscript for publication in the Wind Energy Science Journal.

**2 Specific Comments**

Please provide information on how the WRF simulations have been run, e.g. in batches of how many days, did the authors use spectral nudging, what is the update-frequency of ERA5 data used to provide boundary conditions.

line 40: "largest discrepancies" add percent value as for the neutral and unstable conditions.

line 42: "high-resolution" specify the meaning of this (the cited authors run with 1 km and 500 m resolution).

line 44: "they often fail to capture cluster wake effects across the entire farm, likely due to misrepresentation of internal wake dynamics" I would add that high resolution is certainly good in order to reduce the turbine accumulation within the same cells, with under or over prediction of wake effects, but it breaks the assumptions that are made in most PBL schemes. Even for the 3D PBL scheme, increasing the grid seems to produce even worse results from a wake-evolution perspective. I think this research gap in current literature needs to be mentioned, and it is especially important when looking at wake effects.

line 64: "marine spatial planning" specify in the US.

line 84, 92 (and other parts): the link in the reference Veer, 2023 is broken. This is quite an important reference, as it is cited also later as a report that describes the capabilities of the GPU-accelerated WRF. This reference needs to be accessible and the link to the report needs to be active.

line 110: I don't understand, shouldn't the authors avoid point 4 and only look at the months? So the best January from all Januaries, best February from all Februaries etc? This should automatically provide the year, for each of the selected months.

table 2: months and years are swapped.

line 113: "more meticulous" this is AI-preferred terminology (and also not very meaningful), please change to "alternative".

line 115: "a more complete picture of the meteorological situation is created" this is true only up to a certain extent. The authors are not checking for e.g. statistics of ABL height (not mentioning the geostrophic wind and lapse rate, which also impact power performance). I would arguably say that the picture is complete just enough.

equations 1,2,3: dimensional analysis of these equations is not correct. Also, $P_{ijk}$ is not a power per-se, as it is normalized by density.

line 138: "The thrust coefficient is a dimensionless quantity representing the thrust force" not exactly correct, it rather represent the fraction of local flow momentum applied by the turbine to the flow.

line 158: clearly explain the need to consider peak shaving. Why not use the standard power curves? Do the authors expect peak shaving to have a non-negligible impact on wake deficits? It has already been showed that power output is not much affected, so why this approach given that the focus of this paper is not on loads. Also, layouts are approximated, so I really do not understand the need for this.

line 188: "around a lease are that where there" typo, please correct.

line 188: "wind farm cluster wakes" this study lacks distinction between intra- and extra effects, please see Comment 1.

line 207: "Additionally, we perform this analysis across three different wind lease regions (north, central, and south; Figure 1) to assess regional variations in atmospheric stability" if the authors are interested in stability, a valid choiche could also be to directly look at the mean lapse rate within the ABL (or at the hub height) $\gamma$. This is to rule out possible non-synoptic conditions due to specific atmospheric transients where, even if $Ri >> 1$ (stable) the static stability is neutral or slightly unstable (and vice versa). Also, the relation between $Ri$ and $L$ is of interest. It would be nice to see if the three parameters lead to the same statistics in terms of stable and unstable events.

Figure 5: the authors do not mention, in the entire paper, which curve they used to compute energy losses. This should be explicitly mentioned. The adopted power curve should be added to Fig. 3.

line 268: "the story" change to "these figures".

line 271/272: see Comment 1. Here intra- and extra farm wakes, as well as blockage effects, are being mixed.

line 273: "Large" make lowercase after ":".

line 274/275: "Significant power deficits are more often associated with below-rated wind speeds, regardless of atmospheric stability" this is very obvious and not a novel conclusion of the study. Rephrase to "this is in line with previous studies" or similar.

line 277: "Regions experiencing an annual energy loss of at least 5% are all situated near the wind lease areas. As the threshold lowers to 2%, the affected regions expand further" please try to avoid these kind of comments in a scientific paper. I think it is pretty obvious that wake effects decay moving away from the source. If I misinterpreted, then it was not really clear what the authors wanted to highlight with this comment. I suggest to rephrase or remove it at their convenience.

line 286: please try to avoid the term "wake expansion" which has a very specific meaning in the context of turbine (and also wind farm) wakes. I would suggest using something similar to "affected area", which I think renders the idea much better.

line 288: "CD in determining wake area size and underscores the need to establish an industry-acceptable energy loss threshold to maximize the area available for offshore wind development." this sentence is reductive of the problem, which is complex, in fact higher CD means either more power with equal lease area or the same power with a reduced area. It has to be seen if e.g. the lower $F_{wakearea}$ in low CD regions is compensated by the fact that the lease is larger to effectively produce the same power. There is also a cost problem, both capital and operational, is a larger lease more or less expensive? Installation cost is likely to be higher for a larger area, as vessels have to cover more distance, same goes for operations. So I don't quite get the point of the energy loss threshold, do the authors mean that energy loss has to be lower than a certain percentage outside of the lease by design? Anyways I agree that the wake shadow may be better than wind deficit as a parameter to look at, if calculated between isolated and waked wind farms cases.

table 4: specify that this table considers all atmospheric conditions in the caption (stable,

unstable, below rated etc).

line 297: see previous comment on "wake expansion". Please revise throughout the manuscript.

line 299: change "extend" to "extent", typo.

line 317: numbers (4.92, 3.23) are not matching with table 5, there is something off. Please clarify or correct.

line 341: "Future research should account for these methodological differences and exercise caution..." which formula is more physical in the authors' opinion? It would be nice to know how these formulae, applied to power/energy, compare to actual power/energy deficit calculated from turbine power variable in WRF.

line 362: "and and" typo.

line 376: see previous comment on "wake expansion". Please revise throughout the manuscript.

line 378: "industry-acceptable energy loss threshold to maximize the area available for offshore wind development" first, an analysis of waked and non-waked wind farm is required, to confirm what is the actual delta in loss w.r.t. the intra-wake-only case.

line 384: "accurate" please remove, accuracy was not measured. Energy metric is more "conservative", yes, but it was never demonstrated by the authors how to infer actual power losses from any of these metrics.

line 415: what PC stands for, project capacity? Please define it.

line 420: please define COD, commissioning date?

line 440: please define MA/RI.

line 449: change "specific" to "specifically", typo.

line 463: "Nevertheless, this correspondence is important, as it enables direct comparison of wake areas quantified using both wind speed deficit and energy loss expressed in equivalent percentages" I struggle to relate this sentence with what is stated above.

line 465: "Regarding the distinction between methodologies lies in the normalization

of wind speed, Figure C2 further illustrate using data from Rosencrans et al. (2024)".
Please check this sentence, it does not make much sense.

**References**

Basu, S., Holtslag, A., Caporaso, L., Riccio, A., and Steeneveld, G.: Observational Support for the Stability Dependence of the Bulk Richardson Number across the Stable Boundary Layer, Boundary-Layer Meteorology, 150, 515–523, https://doi.org/10.1007/s10546-013-9878-y, 2014.

---

## Author Comment (AC4)

**Authors' response to Reviewer 1 comments**

Overall, this manuscript is well-written and supported by an appropriate number of references, which demonstrates the authors' thorough engagement with the relevant literature. The use of a GPU-based WRF model is an excellent choice, and the configuration decisions—such as setting the TKE factor to 1 and employing a 1 km mesh resolution at the wind farm—are well-justified and technically sound.

I particularly appreciate the authors' decision to focus on power loss rather than velocity deficit maps, as this approach provides a more meaningful metric for assessing wind farm performance. The AEP loss map presented in Figure 7 is especially informative and visually compelling.

The analysis showing that large velocity deficits at wind speeds above the turbine's rated speed (11 m/s) have a limited impact on overall wind farm production is interesting. While this may not represent a groundbreaking finding, it is still valuable to publish, given the historical reliance on velocity deficit maps in WRF parameterization studies. This contribution helps clarify the limitations of traditional approaches and reinforces the importance of using power-based metrics.

We greatly appreciate the reviewer's positive feedback. We also carefully considered your comments regarding wind farm parameterization and sensitivity testing. Several of these suggestions would require conducting additional simulations. Unfortunately, given our remaining funding and computational resources, conducting all additional experiments is beyond our current capabilities but we did further examine the question regarding the stability classification raised by you and another reviewers by conducting an additional 1-year WRF simulation.

Overall, we still did our best in trying to address most of your comments and we believe this revise manuscript is stronger and more concise than the previous version.

**Major comments**

1- One of the most relevant findings in the manuscript relates to stability differences between the surface and the rotor area. This is an important aspect of wake modeling, and I encourage the authors to expand this analysis. Specifically, could you include a comparison with 1/L at the surface, which is a standard WRF output and widely used in the literature? Additionally, it would be helpful to investigate whether there is a larger mismatch between Ri at the surface and at rotor height near the shoreline. Please also clarify in the text how stability classes were defined using Ri values, as this is currently only indicated in Figure 4 and Table 3.

Thank you for your comment. Reviewer #3 also raised a similar question. Based on your comment, we have conducted additional simulations to extract the Obukhov length (L) to look into this as we did not output the Obukhov length in our initial study. Figure R1 compares $h/L$ (with $h = 10$ m) against Ri calculated for both the near-surface layer (20–50 m) and the rotor layer (20–300 m) at locations within the wind lease region from the newly-conducted simulation. Over the simulated 1-year period, both stability metrics (L and Ri) agree most of the time, and their agreement improves when Ri is evaluated close to the surface (20–50 m). However, even in that case, approximately 14% of the time, L indicates unstable conditions while the near-surface Ri indicates stable conditions. This discrepancy increases to 28% when compared with rotor-layer Ri. Conversely, there are very few data points where L indicates stable conditions and Ri indicates unstable conditions. These results (in particular, the increase from 14% of points in disagreement to 28% of points in disagreement) demonstrate that surface-based stability (L) does not reliably represent stability in

the rotor layer. Because detailed comparison of Richardson number and Obukhov length is not the foucs the scope of this paper, we did not include the corresponding figure in the revised manuscript. However, we have added a discussion on the differences between L and Ri to justify our choice of stability metric.

Lines 252-259: *To examine the difference in stability indication between Obukhov length, L, and Bulk Ri, similar analysis was conducted by comparing $h/L$ (with $h =10$ m) against Ri calculated for both the near-surface layer (20–50 m) and the rotor layer (20–300 m) over the wind lease region. The results suggested that approximately 14% of the time, $h/L$ indicates unstable conditions while the near-surface Ri indicates stable conditions. The discrepancy increases to 28% when compared with rotor-layer Ri. These results further suggest that surface-based stability does not reliably represent stability in the rotor layer. Therefore, for the remainder of the paper, we will use the rotor-layer Ri for further analysis, as it is more appropriate for examining the effects of stability on the wind turbines. However, we note that defining stability in this manner has limitations, particularly when strong local inhomogeneity exists within the rotor layer (e.g., coastal low-level jets).*

Regarding your comments on the difference between Ri at the surface and at rotor height near the shoreline, we conducted the requested analysis and the results are shown in Figure R2. Eight points (colored in red; Figure R2a) are randomly selected to sample the nearshore region adjacent to the wind farm lease area. All points are approximately 5 km away from the shoreline. A comparison between Figure R2b) and Figure 5 in the manuscript shows clear differences: as we move closer to the shoreline, the agreement in stability classification between the surface layer and the rotor layer decreases from 87% to 70%. In addition, the occurrence of cases where the near-surface layer is unstable while the rotor layer remains stable approximately doubles, from 13% to 28%. These results further support our conclusion that surface-based stability metrics do not reliably represent stability in the rotor layer, and that the discrepancy becomes more pronounced nearshore. Although we did not include Figure R2 in the revised manuscript, we have added corresponding text in the discussion to address this point.

Lines 247-251: *Similar analysis is also conducted near the shoreline and the agreement in stability classification between the surface layer and the rotor layer decreases from 86% to 70%. In addition, the occurrence of cases where the near-surface layer is unstable while the rotor layer is stable increases from 13% to 28%. These results demonstrate, as also indicated in Rosencrans et al. (2024), that near-surface stability is not always representative of the deeper rotor layer and the discrepancy is more pronounced nearshore compared to offshore.*

In the revised manuscript, we have added specific text regarding how stability classes were defined using Ri values (Line 240: *In this study, only stable (Ri > 0) and unstable (Ri < 0) conditions are considered*).

[Figure]

Figure R1: **Comparison of h/L (with h = 10 m) and Richardson number calculated for both the near-surface layer (20–50 m; left) and the rotor layer (20–300 m; right) over the wind lease region from the newly-conducted simulation.**

[Figure]

Figure R2: **Difference in near-shore Ri between the near-surface layer (20–50 m) and rotor layer (20–300 m)**

2- The configuration of WRF and the chosen parameterization is appropriate and well-executed. However, the study currently relies on a single model and configuration, which limits its robustness and generalizability. To strengthen the work, I recommend including a comparison with another widely used industry model, such as the Turbopark wake model (for at least one of the wind farms) or the Volker parameterization. This would provide additional context and enhance the credibility of the conclusions.

Thank you for your comment. We agree that adding comparisons with an industry model or an alternative parameterization would further strengthen the robustness of our conclusions. However, as mentioned in the overall response, such efforts are beyond our current resource limitations. Nevertheless, we did conduct an additional simulation and new analysis comparing the Obukhov length with the Richardson number, as this was a common request from both you and another reviewer. In addition, we have added a section on future work to address the uncertainties raised by the reviewer.

Lines 441-449:*There are several uncertainties associated with this study that motivate for future work. First, it would be valuable to apply the same analysis framework using an industry wake model (e.g., the TurboPark model) to assess whether the differences in wake map between wind speed deficit and energy loss approach remain consistent. Second, additional sensitivity studies related to different wind farm parameterization and value of added TKE are needed to quantify the robustness of our results. Finally, a comparison with recently published studies on offshore wind wake assessments demonstrates the complex interplay of multiple factors in determining the final wake area, including the (1) turbine design and layout, (2) simulation techniques, and (3) analysis methods and metrics employed. The variability in the results arising from these factors highlights the need for further research to standardize wake assessment methodologies and develop more robust, universally applicable wake characterization techniques.*

3- Regarding the contour maps presented in Figure 8, which are among the most impactful results for decision-making on wind farm siting: Is it necessary to simulate a full year to generate these maps? A sensitivity analysis using a shorter simulation period would be highly valuable. Furthermore, if a new wind farm were to be added, would it be necessary to rerun the entire cluster simulation, or could the new farm's results be combined with the existing data? Addressing these questions would significantly improve the practical applicability of the study.

Thank you for your comment. For studies involving wake-loss assessment, it is generally necessary to conduct long-duration simulations because accurate characterization of the regional wind statistics (e.g., the wind speed probability density function) is essential. While it is possible in some cases to use a shorter simulation period, that period must be carefully selected to be representative of the regional wind climate. A classic example is Pryor et al. (2021), who employed eleven five-day representative wind-flow scenarios. At the start of this project, we did look into the possibility of running a subset of days that could be representative of the year—however, we found that many days were needed for the probability distributions to converge, and the computational savings were limited. Due to this, we elected to simply run the full TMY.

Lines 110-112:*We considered running only a subset of days to minimize computation time, but found that we could not identify a subset that both suitably captured TMY conditions and significantly reduced the needed computation time, especially when using the GPU-based WRF, so we elected to run an entire year*

If a new wind farm were to be added, it would be necessary to rerun the full simulation for the

entire wind-farm cluster. This is because wakes generated by neighboring wind farms can influence the target wind farm. It is also important to perform a simulation with only the target wind farm included. By comparing these two simulations, one can quantify wake losses associated with both internal and cluster wake effect for the new lease area.

**Minor comments**

1- The improved parameterization proposed by Vollmer et al. (2024) could potentially influence the conclusions of this work. Would it be feasible to conduct a limited test to confirm that the main findings remain consistent under this updated approach?

Thank your for your comment. While we agreed this additional work would strengthen the conclusions of this paper, such effort is beyond our current capabilities. However, this is a potential direction we would like to explore in future work.

Lines 443-444: *Second, additional sensitivity studies related to different wind farm parameterization and value of added TKE are needed to quantify the robustness of our results.*

2- Additionally, please elaborate on the rationale for selecting the three variables used in the TMY methodology. Would including additional variables such as TKE or stability indicators (Ri or 1/L) improve the representativeness of the dataset? Including a standard wind rose plot for the TMY dataset would also enhance the clarity of the analysis.

Thank you for your comment. The 100-m wind speed and direction were selected to provide an accurate representation of wind conditions at turbine-relevant heights from the TMY dataset, as this is critical for the wake loss analysis. The 2-m temperature was included to capture typical surface-level weather conditions. However, we do agree with reviewers that incorporating stability indicators would further improve the representative of the dataset, and correponding text has been added to address this limition. Regarding the wind rose, we did not include it in this revision because the probability distribution comparison between the TMY and ERA5 datasets serves a similar purpose. This comparison demonstrates that the long-term wind climate is well captured in our simulation.

Line129-131: *However, we acknowledge that the current TMY construction is not perfect. Its representation could be further improved by incorporating additional metrics such as ABL height and stability indicators (e.g., air-sea temperature difference, lapse rate)*

3- It would be useful to verify whether a single turbine in the domain reproduces the input power curve when using the parameterization, particularly under stable conditions where low-level jets affect the rotor area. This would help clarify whether some of the observed losses are due to wake effects or to the parameterization's response to vertical wind profiles.

Thank you for your comment. In the current implementation of the Fitch parameterization in WRF, the power output of each $n-$th turbine within the $i, j$ grid cell is only a function of the hub-height wind speed and not the wind profile (Fitch et al., 2012; Skamarock et al., 2019). The power $P_n$ of each turbine in WRF is estimated as follows:

$$P_n(i,j) = \frac{1}{2}\rho U_h^3 \pi (D/2)^2 C_P,$$

where $U_h$ is the hub-height wind speed at the $(i, j)$ grid cell, $D$ is the turbine's rotor diameter, $C_P$ is the power coefficient for $U_h$, and the air density remains fixed at $\rho = 1.23$ kg m$^{-3}$. Note

that in WRF, the power curve is used to estimate the power coefficient as a function of wind speed, which is then interpolated based on $U_h$, and then the turbine's power is estimated following the equation above. Since the power curve is used as an input to the Fitch parameterization to translate wind speed into power, any turbine grid cell in the domain should reproduce the prescribed power curve, regardless of the conditions it is in. The power loss map presented in the manuscript corresponds to a single vertical level (150 m), and the associated power losses shown are due to the reduction in wind speed caused by the total wake effect—both internal and cluster—captured by the parameterization. Redfern et al. (2019) incorporated the rotor-equivalent wind speed into the Fitch parameterization, showing small improvements in power predictions compared to using only hub-height wind speed. Therefore, including the rotor-equivalent wind speed in our simulations is also expected to show margnial improvements.

4- In Figure 6, consider presenting energy loss aggregated at the wind farm level, in addition to the 1 km grid cell resolution. Furthermore, could you explain the energy loss observed far from the wind farms in this figure? Is this related to the well-known WRF parameterization numerical errors in stormy conditions?

Thank you for your comment. In the earlier submission, we did mention the energy loss at the wind-farm level in the manuscript (Line 276: "Under this scenario, power generation for wind farms like Community Wind, Attentive Energy, and Leading Light Wind could experience a greater than 30% reduction in power output."). However, because this information is sensitive and derived from model simulations, we only discussed it briefly. In the revised manuscript, to avoid confusion and at the request of Reviewer 2, we have removed this sentence. However, we have now added a section 3.3 comparing internal and cluster wake losses for one farm (Atlantic Shores South) that specifies the total energy losses for this farm in the presence of wakes.

Regarding the energy loss simulated far from the wind farms (Figure 6b), this feature is mostly due to numerical noise or errors from WRF. Similar behavior is not uncommon in WRF, and Rosencrans et al. (2024) discusses this in detail in Appendix F. In this revision, we have added a sentence to directly point the reader to that paper for further information.

Lines 330-333: *Regarding the energy loss simulated far from the wind farms (Figure 6b), this feature is mostly due to numerical noise or errors from WRF which has been explicitly discussed by Rosencrans et al. (2024) in Appendix F. We encourage the readers to refer to that study for more details.*

5- Finally, for context, are velocity deficit and power loss maps actually used by government agencies for planning purposes? If so, please provide a reference where this application is documented, as this would strengthen the practical relevance of the study.

Yes, velocity deficit and power loss maps are used by U.S. government agencies like the Bureau of Ocean Energy Management (BOEM) and the Department of Energy (DOE) for wind energy planning (Musial et al., 2013a,b). However, we note that the U.S marine spatial planning is a thorough process managed by BOEM, which is responsible for overseeing renewable energy development on the outer continental shelf. As a result, any assessment, including this study, may not fully reflect the actual availability of resources.

Lines 71-73: *We note that the U.S marine spatial planning is a thorough process managed by the Bureau of Ocean Energy Management (BOEM), which is responsible for overseeing renewable energy development on the outer continental shelf. As a result, any assessment, including this*

*study, may not fully reflect the actual availability of resources (Musial et al., 2013a,b).*

**References**

Fitch, A. C., Olson, J. B., Lundquist, J. K., Dudhia, J., Gupta, A. K., Michalakes, J., and Barstad, I.: Local and mesoscale impacts of wind farms as parameterized in a mesoscale NWP model, Monthly Weather Review, 140, 3017–3038, 2012.

Musial, W., Elliott, D., Fields, J., Parker, Z., Scott, G., and Draxl, C.: Assessment of Offshore Wind Energy Leasing Areas for the BOEM Maryland Wind Energy Area, Tech. rep., National Renewable Energy Laboratory (NREL), Golden, CO (United States), 2013a.

Musial, W., Elliott, D., Fields, J., Parker, Z., Scott, G., and Draxl, C.: Assessment of offshore wind energy leasing areas for the BOEM New Jersey wind energy area, Tech. rep., National Renewable Energy Lab.(NREL), Golden, CO (United States), 2013b.

Pryor, S. C., Barthelmie, R. J., and Shepherd, T. J.: Wind power production from very large offshore wind farms, Joule, 5, 2663–2686, 2021.

Redfern, S., Olson, J. B., Lundquist, J. K., and Clack, C. T. M.: Incorporation of the Rotor-Equivalent Wind Speed into the Weather Research and Forecasting Model's Wind Farm Parameterization, Monthly Weather Review, 147, 1029–1046, https://doi.org/10.1175/MWR-D-18-0194.1, 2019.

Rosencrans, D., Lundquist, J. K., Optis, M., Rybchuk, A., Bodini, N., and Rossol, M.: Seasonal variability of wake impacts on US mid-Atlantic offshore wind plant power production, Wind Energy Science, 9, 555–583, 2024.

Skamarock, W. C., Klemp, J. B., Dudhia, J., Gill, D. O., Liu, Z., Berner, J., Wang, W., Powers, J. G., Duda, M. G., Barker, D. M., et al.: A description of the advanced research WRF version 4, NCAR tech. note ncar/tn-556+ str, 145, 2019.